

# Using 3D turbulence-resolving simulations to understand the impact of surface properties on the energy balance of a debris-covered glacier

Pleun NJ Bonekamp[1], Chiel C van Heerwaarden[2], Jakob F Steiner[1] and Walter W Immerzeel[1]

1. Department of Physical Geography, Utrecht University, The Netherlands

2. Meteorology and Air Quality Group, Wageningen University, The Netherlands

*Correspondence to*: Pleun NJ Bonekamp (p.n.j.bonekamp@uu.nl)

**Abstract**

Debris-covered glaciers account for 18% of the total glacier ice volume in High Mountain Asia, however the drivers controlling the melt of these glaciers are still largely unknown and their total contribution to the total glacier melt remains uncertain. Debris influences the surface energy balance and therefore glacier melt by influencing the thermal properties (e.g. albedo, thermal conductivity, roughness) of the glacier surface. In this study, the impact of surface properties of debris on the

spatial distribution of micro meteorological variables, such as the turbulent fluxes, wind fields, moisture and temperature and eventually the conductive heat flux for a debris-covered glacier is investigated.

We simulated a debris-covered glacier (Lirung Glacier, Nepal) at a high-resolution of 1 m with the MicroHH model with boundary conditions retrieved from an automatic weather station (temperature, wind and specific humidity) and UAV flights (digital elevation map and surface temperature), and the model is validated with eddy covariance data. Subsequently, a

sensitivity analysis was performed to ascertain how heterogeneous surface variables control the glacier micro-climate. Additionally, we show ice cliffs are local melt hot spots and that turbulent fluxes and local heat advection amplify spatial heterogeneity on the surface. The high spatial variability of small-scale meteorological variables suggests that point based station observations cannot be simply extrapolated to an entire glacier and should be considered in future studies for a better estimation of glacier melt in High Mountain Asia.

**1. Introduction**

Glaciers in High Mountain Asia (HMA) act as a fresh water supply for millions of people living downstream, and this supply will change due to global warming (Lutz et al., 2013; Wester et al., 2019). Debris-covered glaciers account for 18% of the total glacier ice volume in High Mountain Asia, however the exact melt processes of these glaciers are still unknown and their total contribution to the total glacier melt remains uncertain (Kraaijenbrink et al., 2017).

Debris-covered glacier surfaces differ from clean ice glaciers by surface temperatures that can exceed the melting point considerably, a higher topographic variability and the possibility of a non-saturated surface. As a result debris influences the



surface energy balance and therefore glacier melt by influencing the thermal properties (e.g. albedo, thermal conductivity) of the glacier surface (Reid and Brock, 2010). Glacier ablation is generally enhanced due to the albedo effect by debris

thickness smaller than a few centimetres, while it decreases exponentially with thickening debris by ice insulation (Östrem, 1959).

The energy exchange between the (debris-covered) glacier surface and atmosphere is determined by small-scale meteorological conditions, rather than large-scale weather patterns (Sauter and Galos, 2016). Heterogeneous surface

conditions affect the microclimate resulting in large spatial differences in energy balance components (Reid and Brock, 2010). For example, daytime surface temperatures can range between melting point (ice and water) and 27.5 °C due to inhomogeneous surface heating and variable debris thickness (Kraaijenbrink et al., 2018; Steiner and Pellicciotti, 2016) and the surface roughness ranges from ~0.005 m (gravel) to ~0.5 m (boulders; Miles et al., 2017). Local melt hot spots generally exist on the surface of a debris-covered glacier in the form of ice cliffs and supraglacial ponds (Buri et al., 2016; Miles et al.,

2016), causing highly heterogeneous ablation rates. However, it is not entirely understood how those ice cliffs and ponds form, evolve and disappear. While cut-and-closure of englacial drainage systems are likely an important driver (Benn et al., 2017; Miles et al., 2017b) and the interaction between cliffs and ponds is an important process  (Miles et al., 2017b; Steiner et al., 2019), heterogeneous meteorological forcing over the debris surface likely also plays a role (Buri and Pellicciotti, 2018). The influence of spatial variability and especially with respect to turbulent exchange in the atmosphere however has

so far not been investigated.

Currently there are several methods to model spatial melt of a debris-covered glacier including a multilayer energy balance model (Reid and Brock, 2010) and a fully coupled atmosphere-glacier mass balance model (Collier et al., 2013), where only the latter includes two-way debris-atmosphere feedbacks. However these approaches remain limited in their scope since we

lack insight in the spatial and temporal distribution of surface and meteorological variables, as observations on debris-covered tongues are limited to a few field locations in the Himalaya (Lejeune et al., 2013; Ragettli et al., 2013; Rounce et al., 2015; Steiner et al., 2018), the Karakoram (Mihalcea et al., 2008) and the Tien Shan (Yao et al., 2014), and are all of relatively short time spans in the range of days to multiple months. Extrapolating point measurements remains a challenge on debris-covered glaciers, as measurements from a single weather station are not representative for the complex,

inhomogeneous terrain. Due to this large spatial variation, a high-resolution modelling approach is therefore essential to capture the coupling and interaction between the surface and the atmosphere with sufficient accuracy (Mott et al., 2014).

Turbulent fluxes can play a substantial role in the surface energy balance of a debris-covered glacier (Rounce et al., 2015; Steiner et al., 2018) and are often calculated with the bulk method, where the sensible and latent heat flux are related to the

temperature and moisture gradient between the atmosphere and surface respectively. On debris-covered glaciers however, many assumptions of the bulk-method do not hold due to the high spatial heterogeneity of atmospheric variables, a general





lack of atmospheric stability or inappropriate parametrisations for the complex interaction between the heating surface and the boundary layer (Steiner et al., 2018).

In order to get insight in the microclimate of a debris-covered glacier and therefore wind, humidity and temperature fields, high-resolution turbulent resolving simulations may provide the solution. Large-eddy simulations (LES) studies have been conducted for clean-ice glaciers, focussing on katabatic winds and sensible heat fluxes (e.g. Axelsen & Dop 2009; Axelsen & van Dop 2009; Sauter & Peter Galos 2016). LES often imply a simplification of reality, such as a flat terrain and horizontally homogeneous meteorological conditions (Axelsen and Dop, 2009), though simulations can give insight in

fundamental processes. LES ignore the smallest length scales of turbulence and can be used if the behaviour of those scales can be described as a function of the resolved structures in the simulation. In order to resolve also the smallest length scales direct numerical simulation (DNS) should be used.

Both DNS and LES have advantages and drawbacks for the simulation of atmospheric turbulent flows. Generally, it is

assumed LES represents high Reynolds numbers well, while DNS is only correct if all scales in the flow are resolved. However, as shown by (Moin and Mahesh, 1998), it is in many cases unnecessary to resolve the flow up to the Kolmogorov scale, as many of the statistics of turbulent flows become independent of the Reynolds number at Reynolds numbers far less than the atmospheric one. This is proven for convective boundary layers in the atmosphere (Van Heerwaarden and Mellado, 2016), turbulent channel flow (e.g. Moser et al., 1999; Schultz and Flack, 2013), Ekman flow (Spalart, 2009), and stable

atmospheric boundary layers (Ansorge and Mellado, 2016). Additionally, Dimotakis, (2000) has delivered clear guidelines on estimating whether turbulence is fully developed and we are converging to that range in this study.

Applying LES combined with wall models in complex terrain is questionable, as the Monin-Obukhov Similarity Theory (MOST) that is used to compute the interaction with the wall has been demonstrated invalid already over simple slopes

(Nadeau et al., 2013). Wall modelling on the faces of non-horizontal objects is an unsolved challenge, as all assumptions of the MOST break down. However, no alternative is available, thus MOST is often used nonetheless. This could come with side effects of which the consequences are potentially harder to estimate and interpret than those of moderate Reynolds numbers in the application of DNS.

In this study, the impact of surface properties (roughness, surface temperature and surface moisture) of debris on the spatial distribution of small-scale meteorological variables, such as the turbulent fluxes, wind fields, moisture and temperature is investigated for the Lirung Glacier (Nepal) using a novel DNS model with a spatial resolution of ~1 m. We show the impact of heterogeneous surface conditions and we show how turbulent fluxes are an important contributor to the energy balance of ice cliffs. Observational data are used as boundary conditions, which include a high-resolution DEM (digital elevation map)

and thermal imagery, retrieved from UAV (unmanned aerial vehicle) flights. This is the first high-resolution study for a





debris-covered glacier that investigates the effects of debris on meteorological variables using a turbulent fluxes resolving model. This study improves our process-understanding of debris-glacier melt, and eventually the understanding of the contribution of debris glacier melt to the current river discharge and how this will change in future.

## 2. Methods

*2.1. Study area*

Lirung Glacier is a debris-covered glacier in the Langtang catchment located 50 kilometres North of Kathmandu (Nepal; Figure 1). The Langtang catchment has an area of approximately 560 km$^2$ and is glacierized for 30% and 25% of all glaciers is debris-covered. Lirung Glacier itself is 3.5 km long and on average 500 m wide (Immerzeel et al., 2014a) and ranges in elevation from 4000 m to 7132 m a.s.l.. The surface is highly heterogeneous and debris is composed of a range of textures

from silt to gravel to boulders (Miles, 2017). The average gradient of the tongue is approximately 2 degrees and debris thickness ranges from 0.1 to 2.0 m (McCarthy et al., 2017). This area is influenced during the summer months by the Indian summer monsoon, which provides 70% of the annual precipitation. (Immerzeel et al., 2014b). The winters are relatively dry and precipitation generally occurs only during a few cyclonic events (Bonekamp et al., 2019; Collier and Immerzeel, 2015).

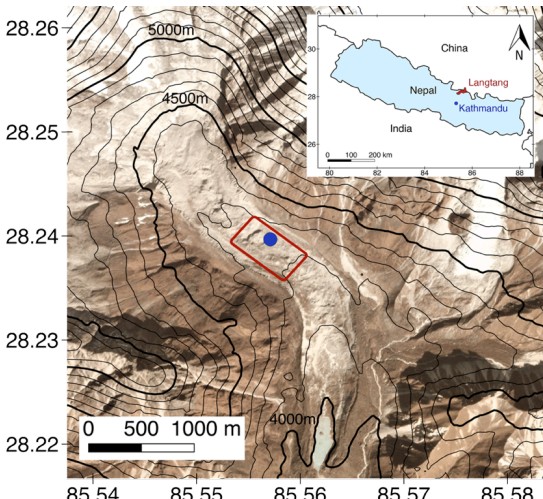

*Figure 1: Lirung Glacier with the microHH domain (red contour) and the location of the AWS (blue point). The background image is a Planet image from 9 December 2018* (Planet

Team, 2017). *The inset shows the Langtang catchment (red) and its location in Nepal.*



## 2.2 Measurements

Data of an AWS on the glacier (Figure 1) is used and observations are made of air temperature, wind speed, relative humidity and incoming and outgoing short- and longwave radiation. Measurements over multiple days at the location exist (see Steiner et al. (2018) for details), however for this study we use the data between 10:30 – 11:30 LT on 12 October 2016

averaged over 10-minute intervals (Steiner et al. 2018). The AWS also included an IRGASON eddy covariance (EC) system measuring high frequency (10 Hz) fluctuations in temperature, humidity and wind speed. Based on this the sensible and latent heat fluxes are calculated as 5-minute averages. The EC system its footprint depends on sensor orientation, wind speed and direction (Steiner et al., 2018), which in combination with an inhomogeneous surface complicates the direct comparison between measurements and simulation. We calculated the footprint area using the EddyPro software and determined the

weighted contribution of each model pixel within the footprint area to the flux observation at the AWS site to make a succinct comparison with the measurements.

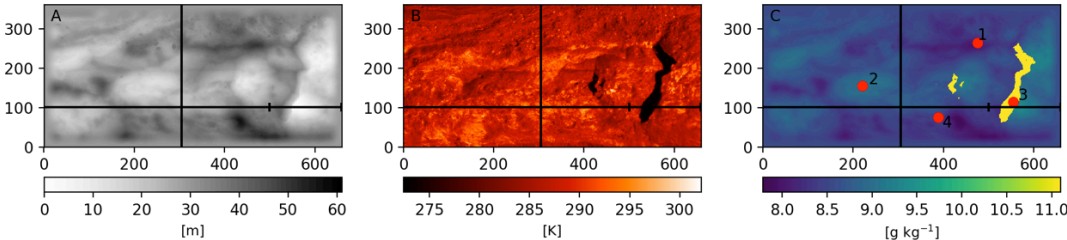

*Figure 2: Boundary conditions used in MicroHH: the DEM (A), surface temperature (B) and surface specific humidity (C). Black lines (A-C) indicate the locations of the vertical cross sections used in Figure 5-7. The vertical cross section used in*

*Figure 9 is a subset of the cross section (x=500-660, y=102) and its start and end point are indicated by small vertical lines. Red points indicate the locations used in Figure 8 (1=dry debris, 2=wet debris, 3=ice cliff and 4=AWS).*

The high resolution DEM is based on the structure-from-motion workflow using optical imagery retrieved on 9 October 2016 (13:00 LT) and is resampled to 1 m resolution for further use (Figure 2A) (Immerzeel et al., 2014a; Kraaijenbrink et

al., 2016). The surface temperature (12 October 2016 11:00 LT) is retrieved with the UAV thermal infrared camera and is biased corrected (Kraaijenbrink et al., 2018). We only use a subset of the UAV data in this research, since the domain is constrained by the intersect of the optical and thermal flight extent, and the domain should be rectangular in the model. The DEM of the domain is detrended, rotated to the main wind direction and smoothened at the boundaries in order to connect the outer left pixels with the outer right pixels of the domain to allow for periodic boundaries. Periodic boundary conditions

presume that the fluxes exiting the domain are used as influx in the next time step. This allows us to investigate processes and feedbacks solely in the domain, as larger forcings are excluded. We only include the glacier surface in the domain and not the surrounding moraines. The final extent of the domain is 660x361 m, the final detrended topography ranges from -





29.4 to 27.7 meter, and the surface temperature ranges from 273.1 K (ice cliff at melting point) to 302.2 K with an average of 282.2K (corresponding potential temperature is 331.3 K). 2% of the domain is covered with ice cliffs and is representative

for the glacier as the ice cliff glacier average is found to be between 1.4 and 3.4% (Steiner et al., 2019).

### 2.3 Model

The MicroHH model (Heerwaarden et al., 2017) is used to simulate turbulent flows in the atmosphere and is an open source direct numerical modeling (DNS) and Large Eddy Simulation (LES) model. We use MicroHH as a DNS model with a constant eddy viscosity, what could be interpreted as a LES with the most primitive eddy viscosity model possible. MicroHH

can be run in parallel and is made for efficient computations. This model allows heterogeneous surface boundary conditions such as topography, surface temperature and surface specific humidity. For specific details of the model we refer to Heerwaarden et al. (2017) but we do give a brief description of the model below.

MicroHH solves the conservation equations of mass, momentum and energy under the Boussinesq approximation and

assumes constant density with altitude what simplifies the governing equations substantially. The conservation of mass is therefore reduced to the conservation of volume in Einstein summation:

$$\frac{\partial u_i}{\partial x_i} = 0,$$
(1)

$u_i$ are components of the velocity vector $(v, v, \omega)$ and $x_i$ the position of the vector $(x, y, z)$. The thermodynamics are a relation between fluctuations of virtual potential temperature $(\vartheta_v')$ and density $(\rho')$ under the Boussinesq approximation by:


$$\frac{\vartheta_v'}{\vartheta_{v0}} = -\frac{\rho'}{\rho_0},$$
(2)

With $\vartheta_{v0}$ the reference virtual potential temperature and $\rho_0$ the reference density. The conservation of momentum is formulated as:

$$\frac{\partial u_i}{\partial t} = -\frac{\partial u_i u_j}{\partial x_j} - \frac{1}{\rho_0}\frac{\partial p'}{\partial x_i} + \delta_{i3}g\frac{\theta_v'}{\theta_{v0}} + v\frac{\partial^2 u_i}{\partial x_j^2} + F_i,$$
(3)

Where $\delta$ is the Kronecker delta, $v$ the kinematic viscosity, g the gravity constant (9.81 ms$^{-2}$) and $F_i$ the external forces originating from for example large scale forcings. We used moist dynamics in our simulations and this implies the liquid water potential temperature $\theta_l$ is the conserved variable in the energy conservation equation.



$$\theta_l \approx \theta - \frac{L_v}{c_p \, \Pi},$$

(4)

with $L_v$ the latent heat of vaporization (2.5 $10^6$ kJ kg$^{-1}$), $c_p$ the specific heat of dry air (1002 kJ kg$^{-1}$) and $\Pi$ the Exner function:

$$\Pi = \left(\frac{p}{p_{oo}}\right)^{R_d/c_p},$$

(5)

with p the actual pressure, $p_{00}$ the reference pressure (1000 hPa) and $R_d$ the gas constant for dry air (287.058 Jkg$^{-1}$K$^{-1}$). The conservation of energy is defined by:


$$\frac{\partial \theta_l}{\partial t} = -\frac{1}{\rho_0}\frac{\partial \rho_{0u_j}\theta_l}{\partial x_j} + \kappa_\theta \frac{\partial^2 \theta_l}{\partial x_j^2} + \frac{\theta_{l0}}{\rho_0 c_p T_0} \, Q \, .$$

(6)

The density of dry air ($\rho_0$) is measured by the eddy covariance system and is set to 0.75 kg kg$^{-1}$, $\kappa_\theta$ is the thermal diffusivity for heat (0.1 m$^2$s$^{-1}$) and Q the external heat source or sink. $T_0$ is the reference temperature profile.

MicroHH gives output of the 3D variables (total specific moisture, liquid potential temperature and the u and v components of the wind) at desired cross sections parallel along the axes x,y,z. The accumulated temperature (thl$_{flux}$) and moisture fluxes (qt$_{flux}$) are given at the surface and converted to a sensible (SHF) and latent heat flux (LHF) by:

$$SHF = \rho \cdot thl_{flux} \cdot c_p \, ,$$

(7)

and

$$LHF = \rho \cdot qt_{flux} \cdot L_v \, ,$$

(8)

where thl$_{flux}$ and qt$_{flux}$ are the diffusive fluxes perpendicular to the surface and are directly dependent on the temperature and moisture gradient between the surface and atmosphere.

### 2.4 Boundary conditions

The bottom boundary condition for the velocity components is set to a Dirichlet no-slip condition (zero velocity at this interface) and the top boundary condition is a Neumann free slip condition (velocity gradient). Random noise is added to the



flow in order to add turbulence and is applied to the wind vectors $u$ and $v$ with an amplitude of 0.1 ms$^{-1}$. We used a constant eddy viscosity of 0.2 m$^2$s$^{-1}$ and a second order spatial discretization scheme is used. A buffer zone of the upper hundred meter is used for numerical stability and the state values decrease exponentially to the top boundary. The boundary

conditions at the DEM surface is a Dirichlet boundary condition and can be prescribed spatially for specific humidity and surface temperature. In our experiments the surface temperature will be set to measured values by the UAV. The specific humidity is not measured spatially by the UAV, although the spatial variability of the relative humidity (RH) is made dependent on the topography to indicate dry higher elevated parts and wetter depressions by:

$$RH = RH_0 - 0.26 \cdot DEM,$$    ( 9 )


with $RH_0$=85% at the lowest point and RH 70% at the highest point of the DEM, with an average of q=8.6 gkg$^{-1}$. This approximation follows the reasoning that melt water entrained in the debris accumulates in depressions. Additionally, finer grained debris equally tends to be found in depressions from wash outs, resulting in a higher retention capacity of the surface. At the location of the AWS the relative humidity (measured at 3.1m) is 66% and with this relationship we assume

the surface is everywhere moister than the atmosphere. The relative humidity during this morning varied from 54% to 100% in time at the AWS location, which is also a typical range during general diurnal variability (Steiner et al., 2018). We assume a spatially constant saturation vapor pressure in the domain based on the air temperature measured by the AWS by Tetens' Formula:

$$e_s = 0.61078 \cdot e^{\frac{17.27 \cdot T}{(T+237.3)}},$$    ( 10 )


and calculate the spatial variable specific humidity by:

$$q = \frac{RH}{100\%} \cdot 0.622 \frac{e_s}{p}.$$    ( 11 )

In order to implement the DEM in MicroHH, ghost cells below the surface are included for interpolation at the surface

following the immersed boundary technique as described by (Tseng and Ferziger, 2003). This method allows for fast computation and senses the presence of the boundary condition by the extrapolated values below the complex surface. The ghost cells itself are excluded from all analysis and are only needed for model performance. The lateral boundary conditions are periodic and this means that air that outflow the domain will enter the domain on the opposite side and act as a lateral boundary condition. The domain can therefore be interpreted as an infinite iteration of the prescribed domain.





**2.5 Vertical profile**

MicroHH is initialized with vertical profiles of liquid potential temperature $\left(\frac{d\theta_l}{dz}\right)$, specific humidity $\left(\frac{dq}{dz}\right)$ and wind $\left(\frac{du}{dz}\right)$. The large-scale pressure force is prescribed by the geostrophic flow components $U_g$ and $V_g$. Profiles of the wind vectors are taken from ERA-INTERIM data at 12:00 UTC, since this profile matched best with the observations and are interpolated in the lowest 100 meter to surface values measured by the AWS. The profiles of the liquid potential temperature and specific humidity are taken constant with height with the value measured at the AWS, since MicroHH was highly sensitive to those initial vertical profiles and varying them did not lead to improvement of the simulation of the latent and sensible heat flux. The ERA-INTERIM profiles contain a temperature and moisture bias at the surface compared to the AWS and in order to get realistic profiles, we interpolated the lower part of the atmosphere to the AWS value. However, this would imply a strong contrast between low air and air at several hundreds of meters and after mixing of the atmosphere, strong gradients and biases appeared in the simulations. We therefore assumed constant profiles for temperature and specific humidity rather than adjusted ERA-INTERIM profiles and that the spin up time (1 hour) is sufficient to acquire temperature and specific humidity profiles that represent the prescribed surface properties.

**2.6 Experiments**

In total 6 experiments are designed to investigate the effects of surface roughness, surface temperature and surface moisture on turbulent fluxes, wind and temperature fields on a debris-covered glacier. The experiments are listed in Table 1. The first two columns indicate the name and the description of the experiments respectively, and the last three columns define which surface boundary conditions are used. If a number is given, this means the surface is homogeneously forced with that value. For the DEM and surface temperature spatially variable measured values are available and this is indicated with *real* in Table 1. A DEM of 0 indicates no topography input is used and the surface is flat and homogeneous. In the last experiment (*REAL*) all variables are prescribed spatially. A specific humidity of 8.6 g kg $^{-1}$ and surface potential temperature of 313.3K are the averages of the measured spatial fields.

Our experiments are representative for the meteorological conditions on 12 October 2016, at 11:00 LT, assuming this is a static state. For each experiment we have output for one hour (without considering spin up) and we consider these results as the range of possible outcomes at 11:00 LT.

The domain extent is 660x331x500 (x,y,z) meter, and 672x384x480 gridpoints are used, so the spatial resolution is approximately 1 meter. The number of grid points is determined by the amount of nodes used on the Cartesius cluster ([www.surfsara.nl](www.surfsara.nl)). One run typically takes 10.5 hours to complete and runs on 1024 processors.



Table 1: Overview of experiments done with MicroHH. The DEM indicates the boundary condition used for the topography (0 means no DEM, ½ DEM is the original DEM halved in height, real is the spatial measured value), $T_s$ the surface potential temperature (313.3K is a homogeneous value, real is the spatial measured value), $q_s$ is the surface specific humidity (8.6 g kg$^{-1}$ is a homogeneously value, the choice for the relative humidity range is described in Sect. 2.4)


| Experiment | Description | DEM | $T_s$ | $q_s$ |
|---|---|---|---|---|
| $HOM_{flat}$ | Homogeneous glacier | 0 | 313.3 K | 8.6 g kg$^{-1}$ |
| $HOM_{1/2DEM}$ | ½ DEM | ½ DEM | 313.3K | 8.6 g kg$^{-1}$ |
| $HOM_{DEM}$ | Roughness effects | Real | 313.3 K | 8.6 g kg$^{-1}$ |
| $HET_T$ | $T_s$ effects 'normal' | Real | real | 8.6 g kg$^{-1}$ |
| $HET_{qdry}$ | $q_s$ dry | Real | 313.3 K | Spatially RH=70-75% |
| $HET_{qmoist}$ | $q_s$ wet | Real | 313.3 K | Spatially RH=70-85% |
| REAL | "Reality" | real | real | Spatially RH=70-85% |

**2.7 Conductive flux**

The surface energy balance determines how much energy is left at the surface and can be used to heat up the debris or melt ice. The conductive flux ($Q_c$) is the energy flux into the debris (Nicholson and Benn, 2006, 2013), and can be quantified by:

$$Q_c = Q_{SW}+Q_{LW}+Q_L + Q_H , \qquad\qquad (12)$$


where $Q_{sw}$ and $Q_{LW}$ are the net shortwave and long wave radiation, $Q_L$ and $Q_H$ the latent and sensible heat flux respectively. $Q_{SW}$ is the sum of the direct incoming ($I_s$), diffuse radiation ($D_s$) and reflected shortwave radiation from surrounding terrain ($D_t$) multiplied by (1-albedo). The surface albedo is taken constant as 0.18 for both the debris as the ice cliffs, as this is measured by the AWS and is in the range of possible albedo for ice cliffs on Lirung Glacier (Steiner et al., 2015). $D_s$ is

calculated as:

$$D_s=V_sk_dI_0, \qquad\qquad (13)$$

with $V_s$ is the sky-view factor, $k_d$ the diffuse fraction and $I_0$ the shortwave radiation measured by the AWS. The reflected shortwave radiation by surrounding terrain is calculated with the albedo ($\alpha$) as:


$$D_t=\alpha I_0(1-V_s). \qquad\qquad (14)$$

$Q_{LW}$ is calculated as:





$$Q_{LW} = V_l LW_{in} + LW_d - LW_{out} \qquad (15)$$

where $V_l$ is the sky-view factor for longwave radiation, $LW_{in}$ the incoming longwave radiation, $LW_d$ The longwave radiation emitted by surrounding debris and $LW_{out}$ the outgoing longwave radiation related to the surface temperature $T_s$:

$$LW_{out} = \varepsilon_d \sigma T_s^4 , \qquad (16)$$

with an emissivity ($\varepsilon_d$) of 0.95 and $\sigma$ the Stefan Boltzmann constant. $LW_{in}$ is taken homogeneous as measured by the AWS
(hourly average) and the longwave radiation emitted by surrounding debris is calculated spatially as:

$$LW_d = V_d \varepsilon_d \sigma T_s^4 , \qquad (17)$$

where $V_d$ is the debris-view factor (see Steiner et al. (2015) for details).

The latent and sensible heat flux are calculated as stated in Sect. 2.3. This method assumes the debris is in steady state and
no heating or cooling of the debris is happening during that period. All fluxes pointed towards the surface are defined as positive. All averages and standard deviations discussed in this paper are spatial averages, unless specified otherwise.

## 3. Results

### 3.1 Spatial distribution of LHF and SHF

In Figure 3 and 4 the average surface turbulent fluxes and spatial variability are shown respectively for all experiments. The
effects of the experiments can be subdivided into effects of surface roughness ($HOM_{flat}$, $HOM_{1/2DEM}$ and $HOM_{DEM}$), spatial temperature ($HET_T$) and surface specific moisture ($HET_{qdry}$ and $HET_{qmoist}$).

### 3.1.1. Surface roughness

The effect of the surface roughness on the SHF and LHF is evident (Figure 3A-F). The turbulent fluxes are intensified with increasing variability in topography, since increasing the surface roughness is directly related to the surface roughness length
and the generation of turbulence. A homogeneous topography results therefore only in small spatial differences of the turbulent fluxes ($HOM_{flat}$, 5 and 2 Wm$^{-2}$ for SHF and LHF respectively). Including a real topography ($HOM_{DEM}$) results in more variation of the turbulent fluxes (64 and 30 Wm$^{-2}$ for SHF and LHF respectively) with lowest fluxes at higher locations of the topography and the highest fluxes in the depressions of the topography. This is caused by the combination of accumulation of heat and moisture in the depressions of the topography and homogeneous surface temperature and specific
humidity, resulting in high temperature and moisture gradients.





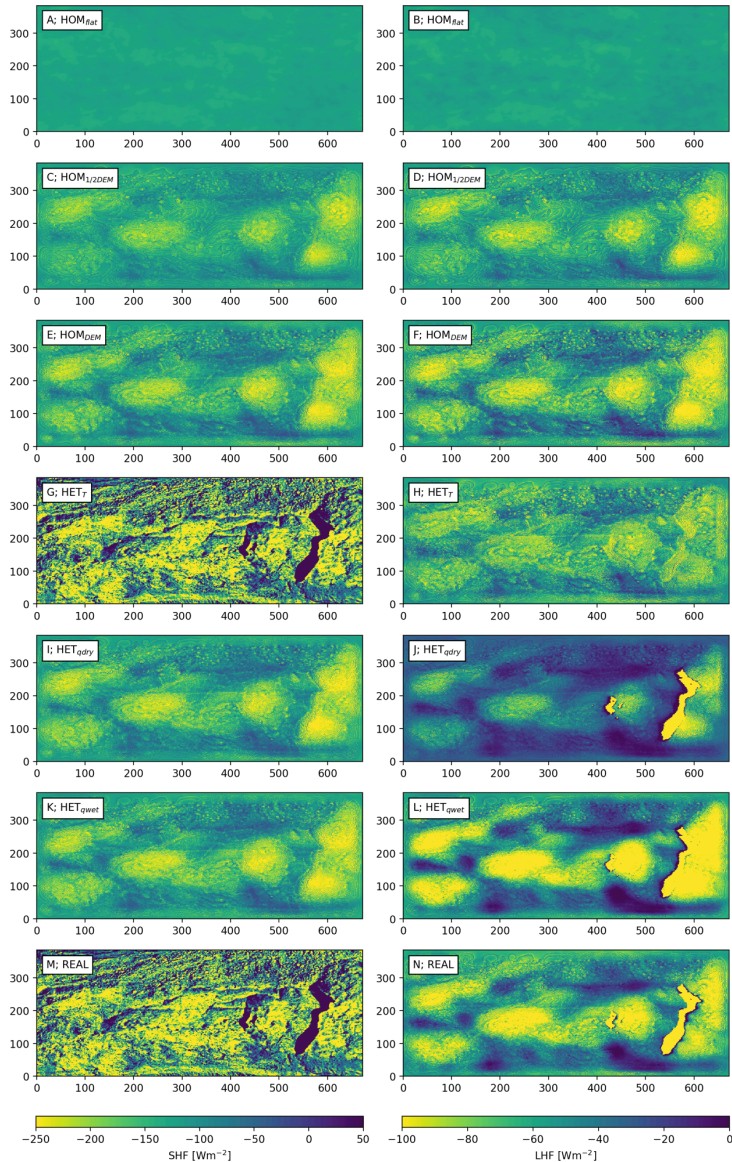

Figure 3: 2D plots of average variables SHF (left panels) and LHF (right panels) for each experiment in the same order as presented in Table 1 (rows). Elevation increases from left to right and the main wind direction is from left to right.





### 3.1.2. Surface temperature

The spatially variable temperature ($HET_T$) has the largest impact on the SHF (Figure 3G). Prescribing the surface temperature heterogeneously ($HET_T$) impacts the surface vertical temperature gradient and is therefore extremely important for the SHF. The LHF is less variable in space when including only spatial heterogeneous surface temperatures. This is because the surface temperature pattern is partly inversely related to the topography and the LHF is mainly driven by the moisture gradient. Cold surfaces are now located at the lowest parts of the domain, where it was warmer in $HOM_{DEM}$ and

LHF is positively related to temperature. Additionally, the spatial variability of the SHF is increased from 64 to 193 Wm$^{-2}$, while the spatial variability in the LHF stayed similar (30 vs 23 Wm$^{-2}$). Due to the heterogeneous temperatures also positive sensible heat fluxes are present at the locations where the surface is colder than the atmosphere. This is particularly important for understanding the energy balance of ice cliffs and ponds (Sect. 3.5).

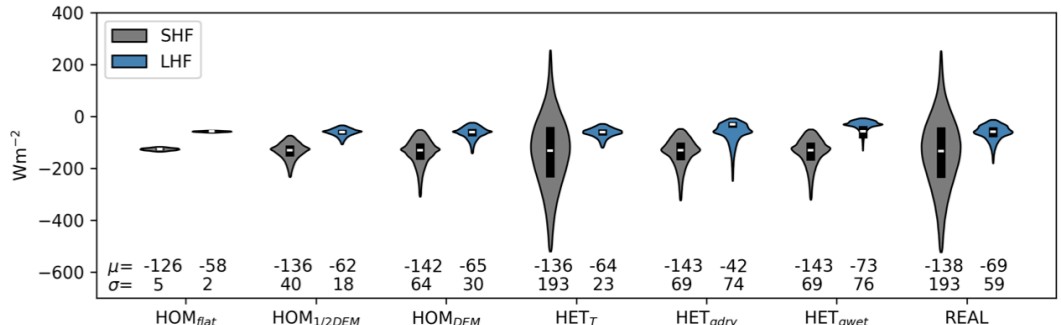

*Figure 4: Violin plots to show spatial variability of time average fluxes (95% confidence interval) in domain for the SHF (grey) and LHF (blue). The numbers indicate the domain average (μ) and standard deviation (σ).*

### 3.1.3. Surface specific humidity

The surface specific humidity has the greatest effect on the LHF. Assuming dry debris ($HET_{qdry}$) results in an average LHF of -42 Wm$^{-2}$, while it is -73 Wm$^{-2}$ for moist debris, indicating the importance of the surface moisture to the LHF. A higher

surface specific humidity results in more evaporation under the same conditions, since it increases the vertical moisture gradient at the surface. Moistening the surface ($HET_{qwet}$) results in less extreme differences spatially in the LHF, since both experiments include saturated areas at the locations where the real surface temperature is 0 degrees Celsius. Moistening the surface will increase the atmospheric specific humidity and since the relative humidity is fixed in both experiments at the cliffs at 100%, the variability in specific humidity decreases. Interesting is the high LHF at the leeward side of the ice cliff

(the main wind flow is from left to right as noted in Figure 3), where wind transports the moisture originating from the ice cliff over the domain.





### 3.1.4. Spatial variation of elevation, surface temperature and specific humidity

Including spatial variation in specific humidity and surface temperature (*REAL*) does not affect the average turbulent fluxes compared to homogeneous conditions ($HOM_{DEM}$) much, however the spatial variability is nearly doubled for the sensible heat flux and tripled for the latent heat flux (Figure 4).

If we assume *REAL* as the truth, the sensible heat flux (latent heat flux) will be underestimated by 9% (8%) when ignoring the topography ($HOM_{flat}$). Assuming both homogeneous surface temperature and specific humidity ($HOM_{DEM}$) results in an overestimation of the SHF of 3% and an underestimation of 4% of the LHF (Figure 4).

Increasing the surface roughness has a larger effect on domain averaged turbulent fluxes than including spatial variable surface temperature or specific humidity. However, prescribing the surface temperature and specific humidity has largest effects on the spatial distribution of the SHF and LHF and results in a high spatial variability. So, for glacier tongue-wide averages, area averaging of the input variables is justifiable but if you are interested in detailed spatial patterns of melt this is questionable.

### 3.2. Vertical distribution of temperature, wind and specific humidity

In Figure 5,6 and 7 cross sections and vertical profiles of all experiments are shown for the wind speed, specific humidity and potential temperature respectively for the lowest 100 meter of the domain. Increasing the surface roughness ($HOM_{1/2DEM}$ and $HOM_{DEM}$) leads to more mixing of heat and moisture in the atmosphere, due to the a higher associated surface roughness length. Close to the surface the DEM has a direct influence on the wind speed; in depressions the wind speed is low and at high-elevated parts the wind speed is higher. Due to the differences in wind speed, the mixing of heat and moisture is also spatially heterogeneous close to the surface. In depressions moisture and heat accumulates and this is mixed into the lower atmosphere by ejections.

The vertical profiles are an average over all model grid points above the surface. The mixing of variables extends higher into the atmosphere when the real topography is included. In the *REAL* experiment, mixing occurs to an altitude of 40-60 meter above the surface, while this is only 20-30 meter in $HOM_{flat}$. On a larger scale it is established that debris influences the near-surface atmosphere (Collier et al., 2015). We show that the micro-scale meteorology is also strongly affected by debris and hence influences for example the local temperature and moisture lapse rates.

The surface roughness causes local differences in wind speeds, especially where there are a lot of elevation differences over a small horizontal range. This is confirmed with station observations, where higher located stations measure consistently higher wind speeds than at lower locations. The three dimensional approach used quantifies the spatial differences in wind





speed and points out the differences with scarce point measurements. In local depressions accumulation of heat and moisture

frequently occurs, and this is further amplified when including heterogeneous surface specific humidity and temperature. If the accumulated heat and moisture are removed regularly by cold and dry air, these heterogeneous differences create local hot spots of the SHF and LHF. Therefore surface roughness plays an important role and can reduce the conductive flux into the debris. At locations where the air is stagnant, fluxes are attenuated since gradients in moisture and temperature between the surface and atmosphere are gradually decreasing by accumulation of heat and moisture. Specifically interesting are the

locations where the specific surface humidity is high and temperature low, such as ice cliffs. We will look more in detail in those supraglacial features in Sect. 3.5.



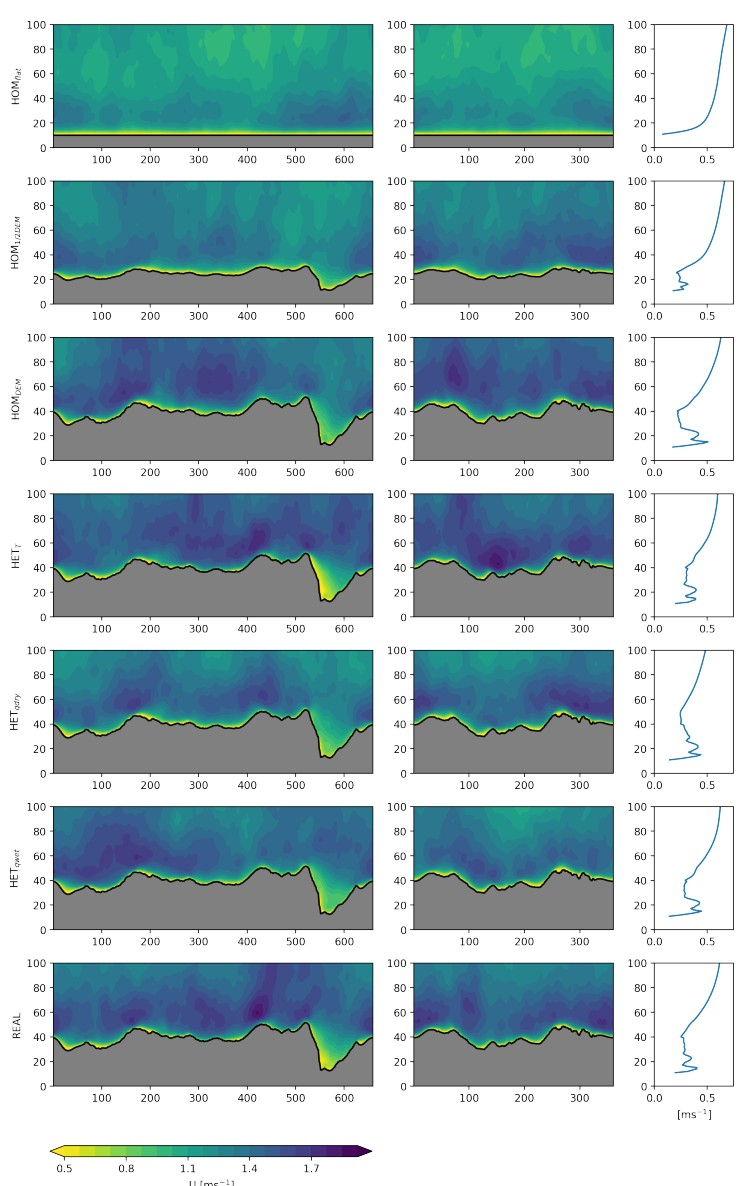

Figure 5: Cross sections (left and middle columns) and average vertical profile (right columns) of the wind speed for all experiments presented in the same order as in Table 1. The location of the cross sections is shown in Figure 2.



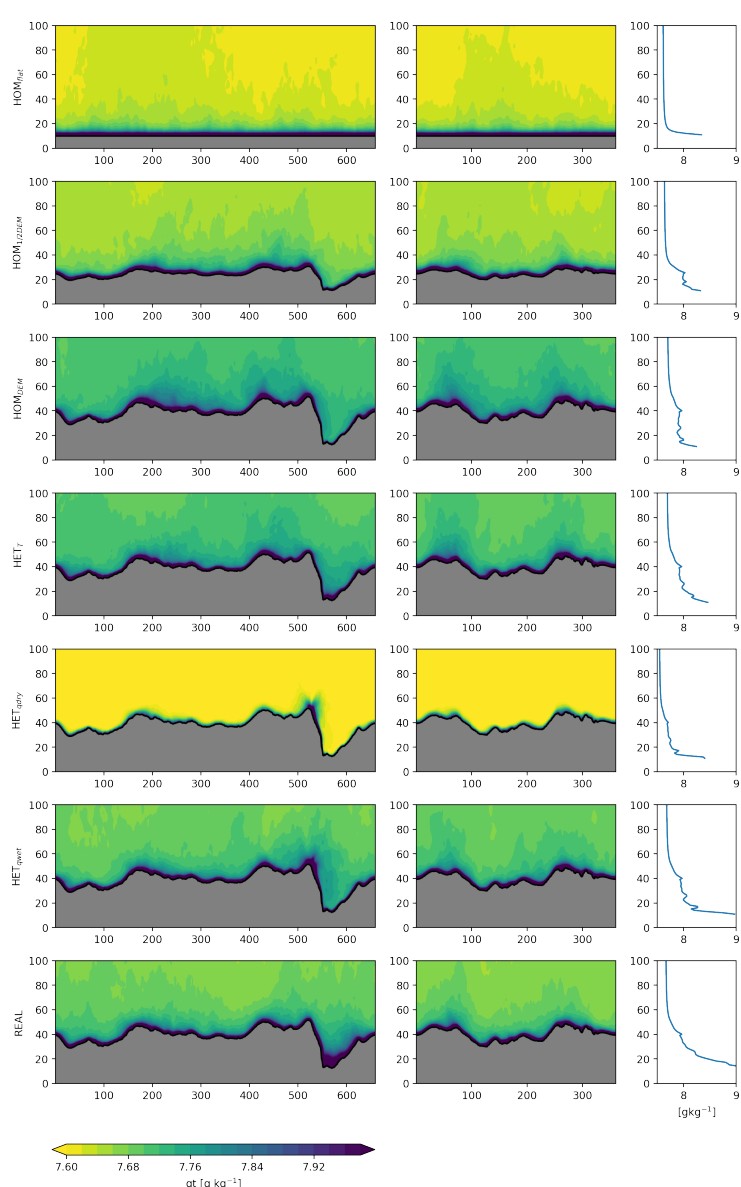

*Figure 6: As Figure 5 but now for the specific humidity.*



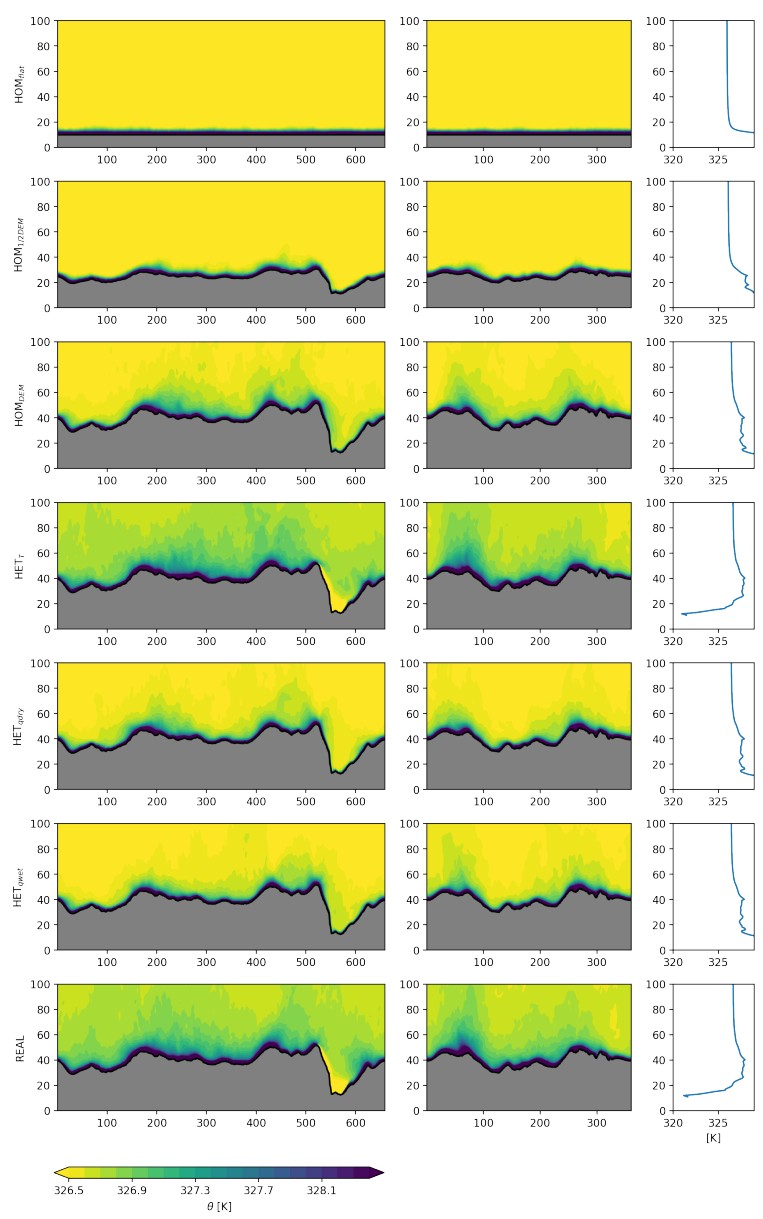


*Figure 7: As Figure 5 but now for the potential temperature.*





### 3.3 Spatial analysis

In Figure 8 the possible range of the SHF and LHF are plotted for four locations: dry debris (A), wet debris (B), an ice cliff (C) and the location of the AWS (D). The exact locations are indicated in Figure 2. The simulations represent a static state at

11:00 LC and we interpret the results as possible range of outcomes of that state. Turbulent fluxes can vary greatly at one location, since they depend on the instantaneous turbulent conditions.

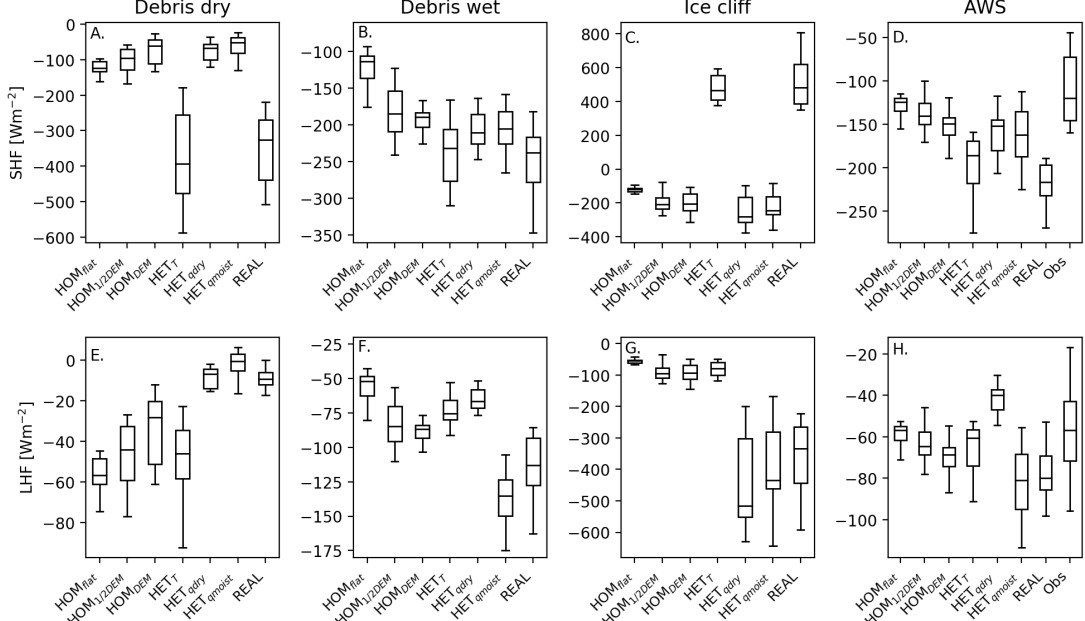

*Figure 8: Box plots to show variability in possible outcomes at four different locations: dry debris (A+E), wet debris (B+F), ice cliff (C+G; all three points are taken as average of 9 grid points) and AWS location (weighted average over the*

*footprint; D+H), for all experiments (Table 1). Observations (Obs) of the SHF and LHF are shown in Figure D and H. For all simulations the last hour of simulation data is taken and is resampled to a 5 minutes average. Measurements are also 5-minute averaged and the time period taken from the AWS is 10:30-11:30 LC.*

### 3.3.1 Debris

For comparison between dry and moist debris, two locations are chosen where both surface temperatures are 291 K in the

*REAL* case. Surface moisture for the dry and wet debris for HET$_{dry}$ are 8.0 and 8.1 gkg$^{-1}$, and for HET$_{qmoist}$ 8.5 and 9.0 gkg$^{-1}$ respectively. Attributing the differences in fluxes between wet and dry debris to surface moisture is not straightforward,





since the spatial surface moisture is dependent on the DEM (dry debris is located at higher elevated parts, while moist debris is located at depressions) and in addition the surrounding grid points influence the turbulent fluxes as well. Dry debris is generally located in areas exposed to higher wind speeds and surface roughness, while the opposite holds for wet debris. The

SHF is more sensitive to surface temperature for dry debris than for wet debris, and in the *REAL* experiment the LHF is approximately 10 times as high at wet debris compared to dry debris. Different sensitivities of turbulent fluxes to surface temperature or moisture are thus applicable in wet and dry climates and the choice of surface boundary conditions should be chosen carefully for simulations.

Domain averaged in the *REAL* case, the LHF at dry debris is lower ($q_s < 8.4$ gkg$^{-1}$, $34 \pm 17$ Wm$^{-2}$) than at wet debris ($q_s > 8.8$ gkg$^{-1}$, $117 + 52$ Wm$^{-2}$) and is caused by less moisture availability. The SHF over wet debris ($-237 \pm 247$ Wm$^{-2}$, excluding ice cliffs) is considerably higher than at dry debris ($-135.3 \pm 161$ Wm$^{-2}$) and is caused by the location of the wet debris in the depressions, where accumulation of heat occurs that increases the vertical temperature gradient as discussed in Sect. 3.2.

### 3.3.2. Ice cliff

The surface conditions at an ice cliff are different than its surrounding debris surface, since the surface is at (or around) melting point and the near-surface air is saturated. As a result, the reversed (positive) SHF is the most pronounced difference compared to a debris surface. Additionally, spatially different surface temperatures result in cold and warm eddies that can pass over the ice cliff and increase the variation in the SHF. The variation in the LHF is mainly influenced by the surface specific humidity, since with heterogeneous surface variables dry and wet eddies can flow over the saturated surface causing

different vertical moisture gradients above the ice cliff. This is described in more detail in Sect. 3.5.

### 3.3.3 AWS measurement comparison

The location of the AWS is included in order to be able to compare the simulations to the measurements. The measurements are an average of the footprint of the station, however, this footprint is varying in time and exceeds the domain slightly. We choose to take a weighted average over the grid points that are located in the domain. The averaged measured SHF is 49%,

lower than the modelled SHF, the LHF is 23% lower. The range between the first and third quartile for the LHF (SHF) is 16 (36) for the *REAL* case and 29 (73) Wm$^{-2}$ for the observations, which shows the model underestimates the variation. The ranges of simulated LHF overlap with the observed range, however for the SHF the observed and simulated ranges do not overlap. The comparison between the simulations and the observations give an indication of the model performance, however a one-to-one comparison is complex, since our simulations are an idealized representation of the reality in a limited

domain and exclude effects from for example the moraines or glacier at higher altitude. The disagreement between simulated and observed fluxes may also be caused by the location of the AWS, which is located close to the moraines. Steiner et al. (2018) estimated the average surface energy balance for Lirung Glacier, at the AWS location to be $\pm 350$ Wm$^{-2}$ for clear-sky



around 11:00 LT, in *REAL* this is 294.2 Wm$^{-2}$, calculated with a weighted average in the footprint of the AWS. However the model domain averaged conductive flux is 348.8 Wm$^{-2}$, indicating the model performs well and within a reasonable range.

**3.4 Surface energy balance**

In Figure 9 the effect of different experiments to the conductive flux into the debris is shown. As input spatial heterogeneous shortwave and longwave radiation fields are used based on the real DEM, and the LHF and SHF are varying depending on the experiments to isolate the effects of the individual experiments. For example, the ice cliff signal in Figure 9A is visible, despite the homogeneous surface conditions in *HOM$_{flat}$* due to the longwave and shortwave signal derived from the real

DEM.

With increasing surface roughness  (*HOM$_{flat}$, HOM$_{1/2DEM}$, HOM$_{DEM}$*) the conductive heat flux becomes more variable, as is also observed for the SHF and LHF separately in Figure 3. Locally it is important to prescribe the surface specific humidity, e.g. for ice cliffs. A wetter environment results in a lower conductive flux (*HET$_{qdry}$ and HET$_{qmoist}$*), while a colder surface

results in a higher conductive flux (*HET$_T$*; Table 2). The effect of the surface temperature (*HET$_T$*) is larger than the surface specific humidity (*HET$_{qmoist}$*) and therefore the conductive flux of *REAL* is closely related to the signal imposed by the surface temperature.

*Table 2, the average conductive flux averaged over the domain, ice cliff cells and debris cells with its standard deviation.*


|  | Domain average (Wm$^{-2}$) | Ice cliff average (Wm$^{-2}$) | Debris average (Wm$^{-2}$) |
|---|---|---|---|
| HOM$_{flat}$ | 391±23 | 674±76 | 363±191 |
| HOM$_{flat}$ | 377±67 | 259±111 | 378±65 |
| HOM$_{flat}$ | 368±102 | 210±187 | 370.8±98 |
| HET$_T$ | 375±205 | 967±193 | 365±190 |
| HET$_{qdry}$ | 389±136 | -142±396 | 399±107 |
| HET$_{qmoist}$ | 358±147 | -81±366 | 366±128 |
| REAL | 368±193 | 674±76 | 364±191 |

A heterogeneous conductive flux contributes to heterogeneous melting, depending on the debris thickness. For example in *REAL* the average conductive flux on the ice cliffs is 674±76 Wm$^{-2}$, while this is only 364+191 Wm$^{-2}$ averaged over the debris. The conductive heat flux at ice cliffs can be used exclusively for ice melt, while for debris the conductive flux is

partly used to penetrate and warm the debris. Turbulent fluxes decrease the energy available for melt by 39% for the *REAL* case averaged over the domain. Over debris, turbulent fluxes reduce the available melt energy by 40% and act as a sink of energy, while on ice cliffs turbulent fluxes enhance it by 51% and are a contributor to melt.





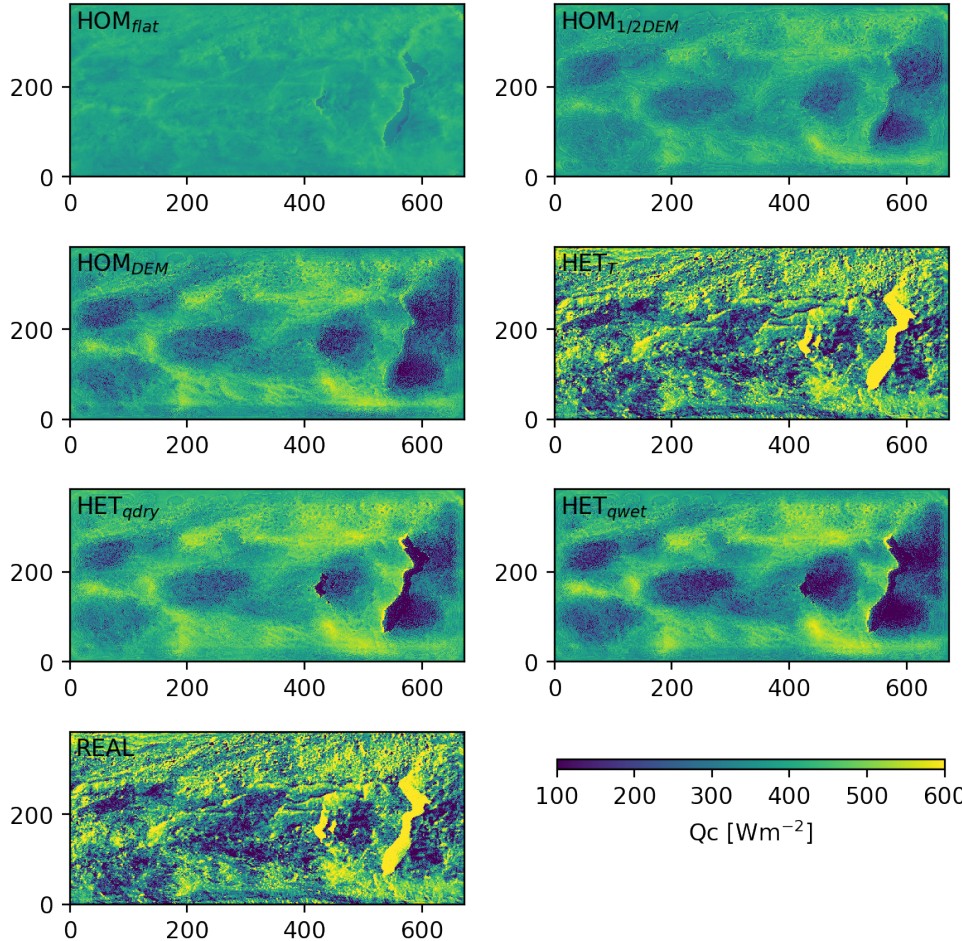

Figure 9: *Effect of turbulent fluxes to conductive flux into the surface for all experiments. A positive flux means energy*
*available to go into surface.*

The surface specific humidity is relatively uncertain and the range used in the experiments is based on scare observation and spatially distributed using a simple relation with elevation. We performed additional sensitivity tests of two extreme cases of a homogeneous surface relative humidity of 20% and of 85%, with the relative humidity at ice cliffs at 100% to quantify the
effect on the conductive heat flux. The domain averaged conductive heat flux when RH=20% is 490±201 Wm$^{-2}$ and 89±276





Wm$^{-2}$ when RH = 85%. This indicates that the mean conductive flux is positive, regardless the surface specific humidity. Second, if the surface relative humidity is lower than the atmosphere the latent heat flux is pointed towards the surface and contributes to the conductive heat flux, while the opposite occurs with a higher relative humidity, what can for example happen at ponds and ice cliffs. Moreover, the conductive heat flux decreases with increasing relative humidity, which shows

the regulating effect of moisture to the total energy budget. At ice cliffs, the latent heat flux can become highly negative if the atmosphere is dry and can even reverse the sign of the conductive heat flux.

In summary, these results show turbulent fluxes can be key in the explanation of the formation of ice cliffs. Locations with already thick debris (and hence higher surface temperatures) do not melt as fast as its surrounding due to the SHF and LHF

that reduce the conductive flux at those places, as well as the general insulation of the debris (Figure 9). A crest will develop introducing the topographic effects as discussed before. This acts as a positive feedback and pronounces the local topographic differences on debris-covered glaciers. These results show that turbulent fluxes can be an additional driver for the typically variable topography of a debris-covered tongue and the formation of ice cliffs besides collapsing channels as hypothesised before (e.g. Benn et al., 2012).

**3.5 Ice cliff analysis**

Ice cliffs on debris-covered glaciers are particularly interesting to study in terms of turbulence given their steep topography, anomalous surface temperature and moisture conditions compared with the surrounding debris. We showed in the previous section that on an irregular surface, local hot spots of the conductive flux into the surface exist, which amplify the melt. Such irregularities likely favour the formation of ever deeper depressions and eventually expose ice cliffs. Our simulations do not

allow dynamic modelling of ice cliff evolution, however we can gain insight in the microclimate around it to better understand the fundamental mechanisms of ice cliff melting. Three dimensional modelling provides therefore insight in processes such as advection of heat and moisture.

In Figure 10 six vertical profiles of wind speed, potential temperature and specific humidity over an ice cliff are shown with

an interval of 10 seconds. On the leeward side of an ice cliff eddies are generated by the thermal and moisture gradients and the topography. In our case the clean ice is on the left side of the local depression and the dominant wind flow is from the left to right. When relatively warm air is advected over the ice cliff this air cools, falls down (first column) and generates an eddy where the accumulated moisture is transported out of the depression and the cold air is replaced by warmer air (columns 2-3). At this point the SHF and LHF are intensified, since the moisture and temperature gradients are increased

within the depression. Intensification of the turbulent fluxes occurs, since warm air from the right side of the depression is transported towards the clean ice by the rotating eddy (column 4). After such an advection event the ice will cool and moisten the air in the depression. The vertical moisture and temperature gradients decrease in time and the SHF and LHF decrease until the process of refreshment repeats itself.



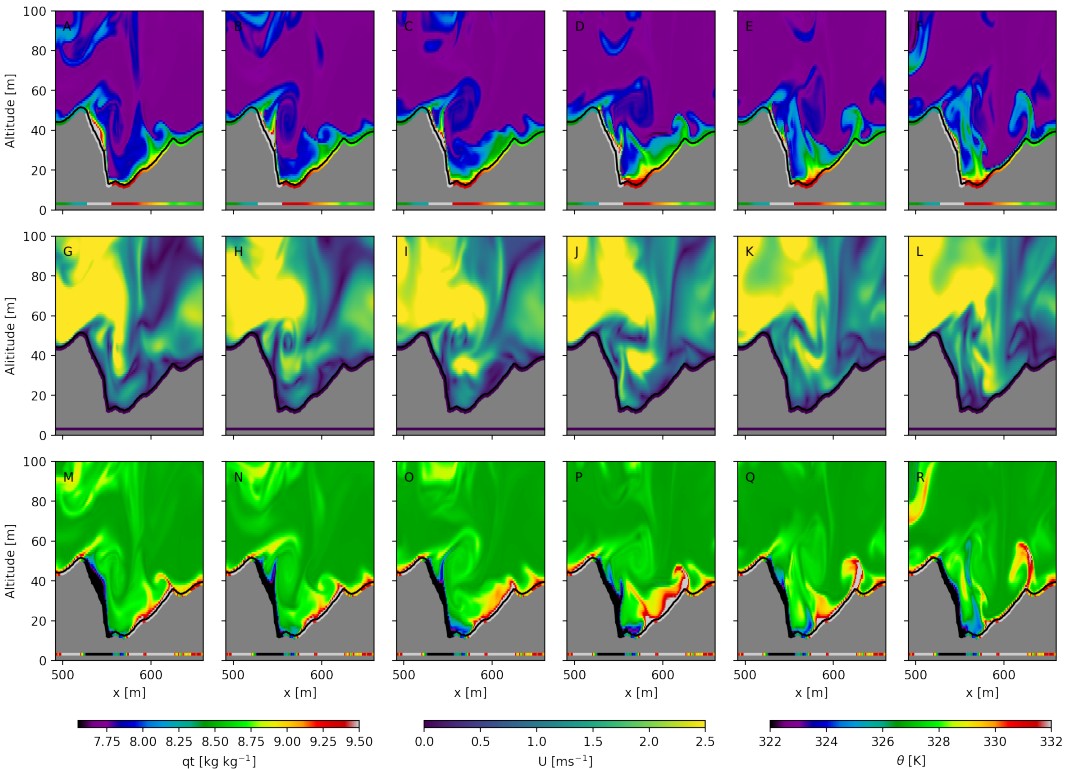

*Figure 10: The specific humidity (A-F), wind speed (G-L) and temperature (M-R) for a zoom-in around an ice cliff (660<x>500m) at time=6570-6630s. The black line indicates the topography. The surface boundary conditions are plotted directly below the surface and for clarity also at y=0.*

During an advection event warm and dry air flows over the ice cliff and is transported into the depression. This causes temporarily a strong gradient and sensible heat contributes to melt, until the air has cooled down. This process gives an intensified melt signal at ice cliffs and causes the cliff to deepen and the ice cliff retreating in this specific case to the left. This is congruent with ice cliff backwasting found in previous studies (Reid and Brock, 2014; Steiner et al., 2015). Similar, Mott et al. (2014) found over snow patches that the net surface radiation is not the only driver of energy exchange between the surface and atmosphere in mountainous areas and that secondary flows induced by surface heterogeneities are also an important driver besides solar radiation. We hypothesize that ice cliff backwasting can also be related to the main wind





direction in the domain. Cliffs that are in the leeward direction of the main wind direction during daytime receive additional warm dry air through the advection mechanism causing extra melt.

Ice cliffs typically have a knick-point shape and is previously explained by differences in local radiation (Sakai et al., 2002),
however this explanation ignores the extra amount of longwave radiation in the depressions (Steiner et al., 2015) and does not explain why crests generally do not flatten in time. Likely turbulent fluxes play an important role in the shape of the ice cliffs as well, by the vertical variations in wind speed induced by the topography. This can be seen in Figure 5, where the wind speed decreases with depth at the ice cliff (x=550m), and the knick-point of the ice cliff is at the location of strongest decline in wind speed. The heat and moisture exchange are therefore strongest at the top of the ice cliff and lowest in the
depression, as the main wind flow does flow over the depression, and does not reach the deepest parts of the ice cliff.

**4. Suitability of DNS**

In order to simulate atmospheric turbulent flows there are two options: DNS and LES, and both options come with advantages and drawbacks. DNS is extremely computational expensive when resolving all scales in the flow and is impossible in our case study. An advantage of DNS is that the correctness of a simulation can be checked by testing the
results of sensitivity tests converge, as we will do in this section. With LES however this approach would not be possible, as sensitivity tests can be evenly wrong caused by the inappropriate surface model under high Reynolds numbers in complex terrain. This means that even when resolving all scales in the flow this does not give direct confidence in the results and LES does not outperform DNS by definition.

To show the turbulence is fully developed in our study, we double the Reynolds number by decreasing the viscosity from 0.2 to 0.11 $m^2s^{-1}$, and we repeated the *REAL* experiment. Computationally the run with low viscosity is 15 times more expensive than the high viscosity as the viscosity is halved and the number of grid points is doubled in x, y and z. The LHF and SHF have the same patterns and are in same order of magnitude at both viscosities (Figure A1). Furthermore, the averaged vertical profiles of potential temperature and specific moisture are also similar, what gives confidence in the use of 1 meter
resolution for this type of simulations (Figure A2).

The energy spectrum of specific humidity is shown in Figure A3; energy spectra of other variables (e.g. temperature and wind) show the same pattern. As this run is very complex and far from idealized, comparison of energy spectra of the viscosities is not straightforward. The energy spectra are derived from horizontal cross section at a fixed height of 65 meter
above the lowest point in the topography, implying that the cross section is not at a constant height above the surface. As a result, the spectrum contains not only a signature of the turbulence but also of the orography. The peak of the spectrum is located around a wave number of 1, what corresponds to a wavelength of $2\pi/1 \approx 6$ meter and is resolved by both experiments.



This shows that large structures are dominant in the flow and small structures are of minor importance. The spectra averages of both experiments differ, though are located in the centre of the bandwidth of one standard deviation.


The simulation at low viscosity naturally resolves smaller scales than the simulation with a higher viscosity, yet the additional variability does not relevantly add variance to the signal and are therefore irrelevant for the flow (Figure A3, wave numbers>11). We therefore conclude a viscosity of 0.2 $m^2s^{-1}$ in combination with a spatial resolution of 1 meter is fine enough to capture the bulk characteristics of the flow as well as its most important features.

**5. Limitations**

The understanding of micro-meteorological processes over debris-covered glaciers with highly variable surface temperature and moisture conditions is limited. In particular the role that turbulent fluxes have in the net conductive energy flux towards the ice below the debris. For the first time, in study, a 3D turbulence resolving high-resolution model is used to gain understanding in these local processes and interactions between the surface and the overlying atmosphere. We show that
such turbulence resolving models provide important insights into these processes, which may support a better quantification of debris covered glacier melt. However, we identify a number of limitations associated to data availability, the model assumptions and computational constraints, which we discuss hereafter.

Information about surface moisture on debris-covered glaciers is scarce, highly variable and very difficult to measure.
However, it is an important variable for the surface energy balance of debris-covered glaciers, since it influences the latent and sensible heat flux. Debris-covered glacier studies normally deal with this unknown by assuming the debris surface is either dry or fully saturated to indicate the range of outcomes (e.g. Rounce et al. 2015). The approach proposed in this study is a step forward for a better representation of surface specific humidity and shows the effect of a partly saturated surface and furthermore a heterogeneous distribution linked to the DEM. In reality the surface moisture is dependent on the surface
material and texture, its relative elevation in the domain and aspect of the location (Qiu et al., 2001). In future, the spatial distribution could also be made dependent on debris grain size rather than the absolute height of the topography to make spatial patterns of surface specific humidity more realistic. During our observation period no large supra-glacial ponds were observed (Figure 1), but surface conditions of ponds can be prescribed in MicroHH in upcoming studies to analyse those as well.


In this study the conductive heat flux is considered, which will be used to gradually warm the debris and eventually melt the ice when the warming front has reached the debris-ice interface. How this energy is partitioned between warming of the the debris and melting depends on the debris thickness, the type of rock, the texture and the moisture content. Future studies can couple a debris energy balance model (e.g. Giese, 2019; Reid and Brock, 2010) to MicroHH and investigate the (timing of



the) total energy reaching the ice, and the impact of debris properties such as thickness, surface moisture and the thermal conductivity.

One of our results is that using spatially variable surface boundary conditions for surface specific humidity and temperature lead to similar domain averages of turbulent surface fluxes compared to a constant surface temperature or specific humidity.

This all relies on the strong assumption that the homogeneous value (in reality taken from a station measurement) is representative for the whole domain. However in reality it remains very hard to locate a station such that it is representative for the whole domain without having prior knowledge of the spatial distribution. This type of high resolution modelling can therefore support the optimal site selection for meteorological observations and ensure its representativeness.

High-resolution modelling is a useful tool to investigate small-scale meteorological processes, due to its explicit treatment of turbulent processes. However, the modelling is still conceptual, since not all processes and feedbacks are included in the model. In this study, only a part of Lirung Glacier is modelled and excludes local influences from the surroundings such as wind flows caused by the lateral moraines and circulations in the valley. Due to the high spatial resolution and computational constraints, the simulation time is now limited to 1 hour and the simulation is stationary; we simulated the domain for 1 hour

assuming it is 1 hour 11:00 LT. Our study provided important insights, but also made clear that turbulence resolving, long-term and transient simulations are currently not feasible. However our results can be used to gain more understanding of surface processes on debris-covered glaciers and obtain parameterizations that can be used in coarser resolution models. In order to understand the small-scale processes at the surface-atmosphere interface on a debris-covered glacier we need high-resolution data and models. A correct representation of debris-covered glaciers would benefit climate, glacier and

hydrological models to give a better estimation of melt water contribution to river discharge.

**6. Conclusions**

The exact melt processes of debris-covered glaciers are largely unknown and their total contribution to the total glacier melt remains uncertain to date. The surface of a debris-covered glacier is complex due to its topography, heterogeneous surface temperature and surface moisture resulting in highly heterogeneous micro-meteorological conditions. In this study, the

impact of surface properties of debris (surface roughness, temperature and specific humidity) on the spatial distribution of small-scale meteorological variables, such as the turbulent fluxes, wind fields, surface specific humidity and temperature for a debris-covered glacier is investigated. This is the first time an in-depth analysis is performed of micro-meteorological variables above a debris-covered glacier with a turbulent resolving model at high resolution (~1 m) and we gained insight in the spatial variability of turbulent fluxes on a debris-covered glacier and what drives these differences.


The surface roughness has the strongest impact on the magnitude of turbulent fluxes and leads to more mixing to higher altitudes due to the higher topographic variability. Surface roughness causes spatial differences in wind speed, with generally





lower wind speeds at lower elevated parts due to isolation, whereby accumulation of heat and moisture is possible. Increasing the surface roughness leads therefore to more pronounced spatial differences in turbulent fluxes.


Heterogeneous surface temperature impacts mainly the SHF, since it influences the temperature gradient between the surface and the atmosphere. A heterogeneous surface specific humidity affects mainly the LHF by influencing the moisture gradient between the surface and the atmosphere. Overall, including heterogeneous conditions lead to higher spatial variability and a larger range of possible outcomes. The variability of the turbulent fluxes can result in a feedback effect that eventually

results in the hummocky terrain typical for debris-covered tongues in the Himalaya and in the extreme case can result in the formation of cliffs and ponds in such depressions where melt was accelerated.

We found that including heterogeneous surface temperature and specific humidity is extremely important when looking at sub-glacier features such as ice-cliffs as those allow both negative and positive turbulent fluxes in the domain. The

microclimate around an ice cliff is influenced strongly by a combination of topographic, surface specific moisture and temperature effects, which favour high sensible and latent heat fluxes. Additionally the progression and persistence of ice cliffs can be dependent on the main wind direction, since at the leeward side of the cliff turbulent fluxes contribute to melt. Longer high-resolution turbulent resolving simulations are needed to investigate fundamentals of small-scale glacier features in detail further.


We showed turbulent fluxes can decrease the energy available for melt at the debris surface by 40% and act as a sink of energy, while on ice cliffs turbulent fluxes enhance it by 51% and are a contributor to melt. In combination with a low albedo this causes ice cliffs to be melt hot spots.

Including homogeneous surface temperature and specific humidity is a good alternative when spatial data is lacking but only when that value is representative for the whole domain and the interest is primarily in domain-averaged outcomes. However, in general the point measurement is not representative for the whole domain resulting in large biases in atmospheric variables when upscaling point measurements to a larger area.

Our results show high-resolution turbulent resolving models can be used to better quantify spatially variable melt and future studies could couple high-resolution models to a full energy balance model to determine the energy reaching the ice. This work is important for glacier mass balance modelling and for the understanding of the evolution of debris-covered glaciers. Subsequently our results can be used to improve the representation of debris-covered glaciers in hydrological and climatological models to determine their contribution to glacier melt.




**Data availability**

The data can be found at 10.5281/zenodo.3375325 and 10.5281/zenodo.3375490.

**Video supplement**

Supporting movies are available at 10.5281/zenodo.3375333

**Author contributions**

PB designed the study together with WI. CH adjusted the MicroHH model for this study. PB performed all numerical

experiments and prepared the manuscript with contributions from all co-authors.

**Competing interests**

The authors declare that they do not have competing interests.

**Acknowledgements**

This project has received funding from the European Research Council (ERC) under the European Union Horizon 2020
Research and Innovation Program (Grant Agreement No. 676819) and Netherlands Organization for Scientific Research
under the Innovational Research Incentives Scheme VIDI (Grant Agreement No. 016.181.308). Supercomputing resources
were financially supported by NWO and provided by SURFsara (www.surfsara.nl) on the Cartesius cluster. We thank Joseph

Shea and Maxime Litt for setting up the measurement station. We are grateful to the trekking agency Glacier Safari Treks
and their staff without whom this work would not have been possible. We thank ICIMOD for their scientific support and
facilitation of the fieldwork. We thank Bart van Stratum for his valuable help with the immersed boundary layer
implementation and the challenges over steep orography.

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



**Appendix A**

This appendix presents the most relevant results of a sensitivity study to the Reynolds number. With this analysis, we demonstrate that the results in the main text do not change when halving the Reynolds number. To provide an intuitive insight into this, Figure A1 shows the computed surface sensible and latent heat fluxes for both simulations. This figure clearly demonstrates nearly identical patterns and magnitudes of the surface fluxes. The similarity can also be understood from scaling arguments. Over a rough surface, the wind-driven surface fluxes can be approximated as:

$$LHF = \rho \cdot L_v \cdot c_{dq} \cdot (u - u_0) \cdot (q - q_0),$$
$$SHF = \rho \cdot c_p \cdot c_{dh} \cdot (u - u_0) \cdot (T - T_0),$$

where $\rho$ is the air density, $c_p$ specific heat, $C_{dh}$ and $C_{dq}$ the exchange coefficient for heat and moisture respectively, u the wind speed, $u_0$ the wind speed at the surface, T the temperature and $T_0$ the temperature at the surface, $L_v$ the latent heat of vaporization, q the specific humidity and $q_0$ the specific humidity at the surface. In our case, both simulations have the same

boundary conditions for temperature and humidity, and resolve nearly identical atmospheric fields (Figure A2). As atmospheric profiles are identical, the only place where low Reynolds number effects could manifest itself is via $C_{dh}$ and $C_{dq}$, and therefore in the magnitude of the surface fluxes, yet Figure A1 shows that this is not the case.

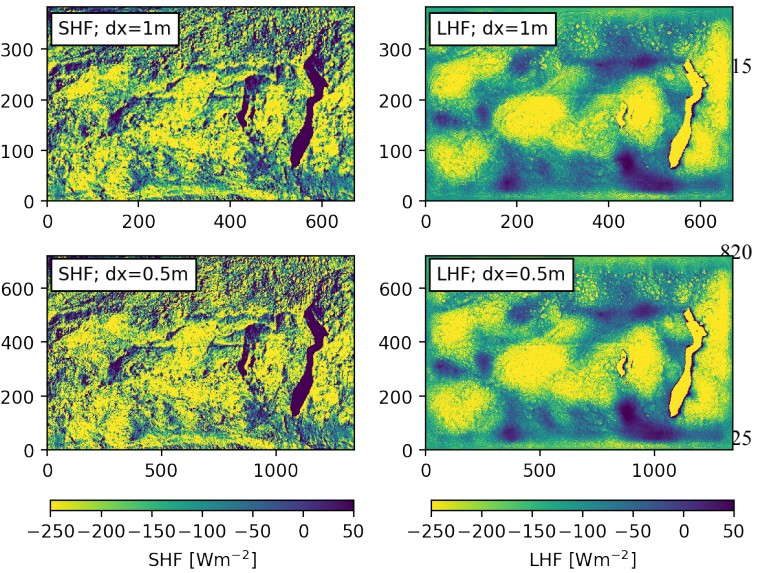

Figure A1: The averaged sensible (left panels) and latent heat flux (right panels) for the *REAL* experiment with a viscosity of 0.2 m²s⁻¹ (dx=1m; upper panels) and with viscosity of 0.11 m²s⁻¹ (dx=0.5m; lower panels).



Further proof of the independence of the bulk quantities (profiles of means and variances) can be found in the streamwise spectra of specific humidity (Figure A3). The additional variance that is resolved in wavenumbers >11 does not relevantly contribute to the total amount of variance. This can be visually inferred from Figure A3, as the total variance is the area

under the graph, and the newly added variance is invisible to the eye.

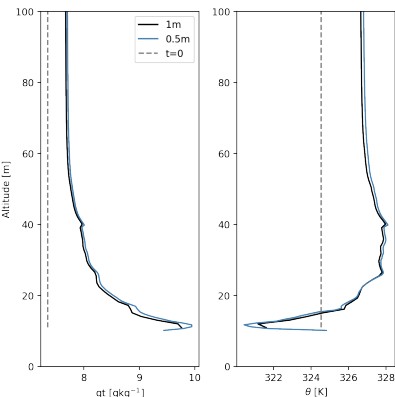

Figure A2: Total specific humidity (left) potential temperature (right) profiles for the REAL experiment with dx=1m (black), dx=0.5m (blue) averaged over the simulation hour and their initial profiles at t=0 (gray).

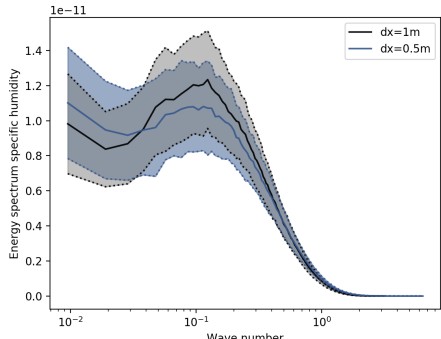

Figure A3: Energy spectrum for the total specific humidity at 65 meter above lowest point of the topography, averaged over the simulation hour. Shading indicates one standard deviation. The vertical axis is pre-multiplied with the wavenumber.