# Peer review of "Using 3D turbulence-resolving simulations to understand the impact of surface properties on the energy balance of a debris-covered glacier"

_The Cryosphere, 2019_

## Referee Comment (RC1) · Anonymous Referee #1 · 13 Dec 2019

This paper conducted a comprehensive study of energy balance for a debris-covered glacier using turbulence-resolving numerical simulations. The paper is well organized and written, and the conclusion is supported by the results. I am generally favorable to publication. However, I feel that the following point needs to be addressed before publication.

My major concern is that the one-meter spatial resolution used in the paper is way beyond the dissipation range, so I would be very hesitated to call it a DNS. I would like to be clear that I am not questioning the validity of the numerical results and the corresponding conclusion in this paper. I agree that a one-meter resolution, constant eddy viscosity model should outperform a 10-meter resolution LES-SGS model. However, instead of calling it a DNS, I would suggest calling it a different name, e.g., a constant eddy viscosity LES, a high-resolution LES, or a quasi-DNS? The authors did show that doubling the spatial resolution (and the Reynolds number) had a small impact on the simulation results. In my opinion, this only indicates that, numerically, increasing the grid resolution has a limited impact on the simulation results. However, whether a one-meter resolution simulation can explicitly capture the right physics (small-scale turbulence behavior) in the atmosphere is still not quite clear. I suggest the authors conduct a validation that compares the quasi-DNS results with field experiments. I am also curious to see how a LES model behavior, compared with the quasi-DNS runs. Without such a comparison, it is hard to justify the one-meter resolution DNS configuration.

Some minor comments:

Line 97, page 3. "using a novel DNS mode", please be more specific in terms of what novel algorithms or techniques are used in the DNS solver?

Page 8. I suggest the authors add a schematic to illustrate the boundary conditions and especially, the immersed boundary method used to represent the DEM. Also, please be more specific why did the authors use 0.2 m^2 s^-1 as the eddy viscosity, as opposed to other possible values?
* * *

---

## Referee Comment (RC2) · Anonymous Referee #2 · 2 Jan 2020

Overview:
- This is an important study that sheds light on the small scale variations in turbulent heat fluxes across the surface of a debris covered glacier using a high-resolution computational fluid dynamics model applied to the near surface atmosphere. It uses a novel approach to provide valuable insight into the relative importance of key, typically measured meteorological variables and, I believe, will be of great interest to the debris covered glacier scientific community (particularly to distributed modelers).
- The author/s clearly has/ve a solid understanding of the physical processes underlying the stated observations in the simulation results. I think this strength could be highlighted more by, in a renamed Results and Discussion section, explaining the physical processes first, then how the results demonstrate them.
- The authors show that variations in melt over the surface of a debris covered glacier is due not only to debris thickness but also to variations in turbulent fluxes. Perhaps this is intuitive, but this study is the first to show it by simulating spatial patterns in wind, humidity, and temperature. This paper shows that turbulent fluxes are important to understanding the development of ice cliffs, which have previously been shown to be ablation hotspots. It importantly shows that surface heterogeneities are an important driver of energy exchange.

General comments:
- The discussion of the study's focus on micro meteorological variables must be honed. Turbulent fluxes are determined by wind speed and surface roughness length, as well as by temperature (sensible) and humidity (latent). In the abstract, the manuscript reads "turbulent fluxes, wind fields, moisture and temperature…"; in the conclusion, the manuscript reads "turbulent fluxes, wind fields, surface specific humidity and temperature for a debris-covered glacier is investigated." I suggest reframing the language around the purpose of this study: specifically, not listing (and, thereby, implying) turbulent fluxes as separate from wind, humidity, and temperature.
- This manuscript needs English editing (grammar and punctuation) beyond what can be provided in my review; I made some suggestions, but the manuscript needs major editing for readability. The English hampered my comprehension of the scientific basis of the paper.
- I found the Introduction especially confusing to follow–partly because of wording choices (e.g. however, thus, and nonetheless in a single sentence) and partly because there is insufficient detail on key elements of an introduction but superfluous detail on non-essential inclusions (e.g. methods and wall modeling). The authors do not describe LES and DNS beyond spelling out the acronyms, and the authors do not

discuss the reasoning behind a spatial resolution of ∼1 m. The section needs clearer language to communicate a revised structure of problem/question –> hypothesis –> aims (generally exploring turbulent fluxes) –> objectives (specific, describing methods). I find that the statements of the problem (incomplete understanding of the drivers of heterogeneous melt patterns) and hypothesis are roundabout and unclear. In the last paragraph of the introduction, the aim and objectives are intermixed.

- Simulations: I think the suite of simulations provides valuable insight into the different variables in the energy balance. However, the author needs to make a distinction between humidity  moisture and surface roughness  topography/DEM as well as improve the explanation for the source of the distributed temperature and humidity data. A priori, it seems that it could be useful to conduct simulations with halved temperature and humidity deviations from the means. I think that the justification and explanation for choosing the 7 simulations needs to be strengthened and clarified, as I miss the reasons for performing the specific simulations.

Specific comments:

- Title: suggest simplifying to "using 3D turbulence-resolving simulations to investigate the energy balance of a debris-covered glacier"
- Abstract: needs to be original and not contain exact sentences from the body of the manuscript. It would be appropriate to mention that you designed a series of simulations that differed in input parameters in order to isolate and investigate the effects of varying those parameters.
- Line 12: remove first "total"
- Line 20: suggest replace "ascertain" with "provide insight into"
- Line 32: explain/rephrase "non-saturated surface" or provide a reference
- Line 43: add a citation for gravel. Be more specific that you are talking about surface roughness lengths. Later, you use surface roughness interchangeably with topography (and DEM). Specifying length here would eliminate subsequent confusion.
- Line 51: what does "spatial melt" mean? You don't cite any distributed energy

balance model on debris covered glaciers: e.g., Reid et al (2012), Fyffe et al (2014)
- Line 54: "we" or "they"?
- Line 55: there are many remotely-sensed observations. If excluding these, be specific. Also, missing references with data: Vincent et al (2016), Nicholson et al (2018), Nicholson Mertes (2017)
- Line 58: this is a bold claim. Be specific for what observations are over short time spans.
- Line 60: modeling would lend significant new insights, but you haven't argued it is "essential." Language implies it is the only method to shed light on the question, whereas it is only one approach.
- Line 64: "heat fluxES"
- Line 65: "gradientS"
- Line 65: Steiner et al (2018) found that bulk methods overestimate turbulent heat fluxes. . . seems relevant to mention
- Line 66: summarize the "many assumptions" since this point is central to the problem you aim to address
- Line 70: "therefore" is for results, not for clarification. Suggest id est here. "and therefore wind, humidity and temperature fields" –> "(i.e. its wind, humidity and temperature fields)"
- Line 86: "we are converging to that range in this study": meaning unclear
- Line 105: inconsistent formatting
- Lines 107 - 108: rephrase sentence
- Figure 1 and most subsequent figures: include axis units and labels!
- Line 131: suggest section title "field measurements." Section as a whole needs tightening of language to be more to-the-point. It is difficult to decipher meaning.
- Line 135: what is the purpose of this citation?
- Line 149: what is the purpose of these citations? Consider adding the words "following" or "after" if that's what you mean
- Lines 151 - 152: info in sentence "we only. . . the model" needs to be added to the

previous section to explain the extent of the microHH domain

- Line 157: if this dash is to indicate negative, make sure it is on the same line (and page!) as the following number... and that Fig2A has the stated range included in its colorbar.
- Line 159: suggest rewrite "2- Line 162: suggest replace "used... LES)" with "a computational fluid dynamics model designed to simulate turbulent flows in the atmosphere through direct numerical simulation (DNS) and large eddy simulation (LES)."
- Line 164: suggest replacing "what could be interpreted as" with "which effectively renders it"
- Line 166: refer the reader?
- Line 170: instead of what, which (error appears many times)
- Line 171: instead of therefore, thereby
- Line 173: instead of are, is
- Lines 193 - 194: these lines need review with respect to units and consistency. (Density is not kg/kg; what is "thermal diffusivity for heat" with a value of 0.1 $m^2/s$? If you mean thermal diffusivity of water, give a calculation with specified T and P. Should be $\sim$0.1 x $10^{-7}$ $m^2/s$)
- Line 196: "accumulated temperature" is not intuitive. Please explain.
- Line 209: here and elsewhere, meterS when more than 1
- Line 210: condition is
- Line 232: themselves
- Line 233: suggest "are periodic, such that air flowing out of one side of the domain will enter on the opposite side."
- Line 249: the table lists seven experiments, not six
- Lines 263 - 264: these sentences are superfluous
- Line 265: By this point, I am missing an in-depth description of experiment design and what question each experiment was designed to answer. A reader can possibly deduce this from the results, but the purpose should be stated more explicitly.

- Table 1 caption needs proofreading (homogenous or homogenously?; should be "spatially varying")
- Line 273: 2012 (not 2013)
- Line 300: suggest "all fluxes are defined as positive towards the surface except for the conductive heat flux"
- Line 302: suggest renaming the section "results and discussion" and including more discussion rather than assuming that the reader can deduce the significance from the figures (e.g. line 308: "the effect of the surface roughness on the SHF and LHF is evident (figure 3A - F)"). This section would benefit from an overview of the fact that the authors perform seven experiments designed to very key parameters that control turbulent heat fluxes in order to investigate the relative importance of various controls. Then, the subsections and figure captions could be strengthened by statements of which tests were designed to test which variables.
- Line 306: introduce that for temperature you contrast $\mathrm{HET}_T$ with $\mathrm{HOM}_{DEM}$ and mention how you plan to incorporate "REAL"
- Figure 4: this figure summarizes the results of the experiments very well. The caption could benefit from a reminder of the sign convention for fluxes.
- Section 3.1.4: it is not clear why only three of the experiments are discussed in this section. The last sentence of this section raises a very important point, which should be discussed further.
- Line 351: "spatial variations in"
- Figures 5, 6, 7: what kind of cross sections are these? (Reference the black lines in figure 2 in the manuscript text around figures 5 - 7.)
- Line 360: use topography rather than DEM. The two are not interchangeable.
- Line 364: "ejections." What about diffusion and advection?
- Line 377: "reduce," but can't it also increase? Suggest "alter."
- Figure 7: the last column of some figures is striking because the change with height (cold to warm versus warm to cold) differs between experiments. Must discuss this and other features of the figures.

- Line 372: point out specific features in figures
- Lines 395 and 402: LC or LT?
- Figure 8: dry debris and wet debris (change word order). It would be more intuitive to group the dried debris as A and B, the wet debris as C and D, etc.
- Lines 395 - 396: this sentence seems to negate the importance of the following figure. Needs clarification.
- Line 405: paraphrase "REAL case" for clarity
- Line 407: "spatial distribution of surface moisture..."
- Line 410: good insight into physical properties. Not clear how to disentangle effects of topography from effects of debris.
- Lines 411 - 413: this is an important point which is difficult to discern because of the language. Rewrite. Also, author(s) need to distinguish their contributions from physical principles. SHF more sensitive to $T_{surf}$ for dry debris is true b/c water has a higher heat capacity than air. "10 times as high" in next line is something learned through the model.
- Lines 416 - 417: If these are averages, why is there an uncertainty? And why is it greater than the average?
- Line 421: "near-surface air is saturated" contradicted at end of paragraph
- Line 429: weighted how? Approach needs an explanation.
- Line 441: Suggest "figure 9 shows conductive flux into the debris under the seven simulations."
- Line 452: can you say anything here about the physics with respect to thermal conductivity, density, and heat capacity? Conductive heat flux is determined by the temperature gradient in the debris, so this is expected. Clarify what additional insight your simulations provide.
- Table 2 line 480: what is the significance of the non-normal distributions, for which standard deviations exceed the means? Consider using other statistical metrics instead/also.
- Table 2: refer to this table in the text

- Table 2: what is the breakdown of ice-cliff area vs. debris area in the domain? Mention in caption and discuss in text if you contrast in a table.
- Lines 467 - 472: relate to recently published findings on ice cliffs in HMA
- Line 477: observationS of what value(s)?
- Lines 504 - 505: clarify with "every 10 seconds of [time interval] in the simulation"
- Figure 10: label ice cliff face, give variables in headings for each row of figures (A - F, G - L, and M - R), and specify the exact time interval in the caption. Also, should be $660<x<500$.
- Line 529: "and is" to "that was"
- Line 531: turbulent fluxes likely play
- Line 532: it is appropriate to refer to the figure, but the reader cannot "see" the windspeed derivative where the ice cliff changes slope. Labeling the ice cliff and circling the region of interest on the figure would help.
- Line 535: "wind flow does flow": rewrite
- Section 4: suggest renaming as "sensitivity to Reynolds number" and start with a short description of the Reynolds number and why you chose to perform a sensitivity test on it. The first paragraph of this section states that both DNS and LES are impossible. The meaning of this paragraph is especially difficult to discern from the English that is used.
- Lines 545 - 550: what is the effect of the different resolutions on the profiles near the surface, where the difference is most apparent? You show the resolution is not too large for achieving accuracy, but could the same patterns be captured with a resolution larger than one meter? How much larger?
- Lines 561 - 563: rephrase
- Line 556: topography, not orography
- Section 5 "Limitations": this section especially needs proofreading by the author(s). The writing makes it difficult to discern many of the concepts, which are ones important to the paper.
- Line 576: add that debris moisture is important to not only turbulent heat fluxes but

also the conductive heat flux that ultimately melts glacier ice

- In this discussion of moisture, can you add some discussion of season and any implications of your findings for the monsoon season in particular?
- Line 598: here, it sounds like the AWS station you used *happened* to be in a spatially-representative place but that you got lucky because "in reality it remains very hard to locate a station such that it is representative for the whole domain." If this is not the case, please change the language. Additionally, it would be helpful if you quantified the amount of bias that could be introduced by upscaling point measurements not representative of the domain.
- Line 615 - 618: these lines read that you investigated the impact surface temperature and specific humidity have on surface specific humidity and temperature. Please rewrite with greater clarity.
- Line 642: I would think that that the bare ice on ice cliffs has a higher albedo than debris covered surroundings; explain or cite otherwise.
- Line 648: this paragraph is important to include only if you quantify and show an example of the large biases that are possible.
- Line 661: add labels to axes in the .gif's (videos)
- References: cite publications in The Cryosphere, not The Cryosphere Discussions where possible (e.g., Rounce et al., 2015)

---

## Author Comment (AC1) · 3 Feb 2020

**Reply to referee 1**

This paper conducted a comprehensive study of energy balance for a debris-covered glacier using turbulence-resolving numerical simulations. The paper is well organized and written, and the conclusion is supported by the results. I am generally favorable to publication. However, I feel that the following point needs to be addressed before publication.

My major concern is that the one-meter spatial resolution used in the paper is way beyond the dissipation range, so I would be very hesitated to call it a DNS. I would like to be clear that I am not questioning the validity of the numerical results and the corresponding conclusion in this paper. I agree that a one-meter resolution, constant eddy viscosity model should outperform a 10-meter resolution LES-SGS model. However, instead of calling it a DNS, I would suggest calling it a different name, e.g., a constant eddy viscosity LES, a high-resolution LES, or a quasi-DNS? The authors did show that doubling the spatial resolution (and the Reynolds number) had a small impact on the simulation results. In my opinion, this only indicates that, numerically, increasing the grid resolution has a limited impact on the simulation results. However, whether a one-meter resolution simulation can explicitly capture the right physics (small-scale turbulence behavior) in the atmosphere is still not quite clear. I suggest the authors conduct a validation that compares the quasi-DNS results with field experiments. I am also curious to see how a LES model behavior, compared with the quasi-DNS runs. Without such a comparison, it is hard to justify the one-meter resolution DNS configuration.

*We thank the anonymous reviewer for their time, and their insightful and helpful suggestions to make the manuscript more accessible and accurate.*

*To make the differences and links between LES, DNS more clear we have now included in the introduction L97-102:*

*"Turbulence can be simulated by two techniques: LES (Large Eddy simulations) and DNS (Direct Numerical simulations). The difference between them is the treatment of the smallest scales in the flow; LES uses a subgrid parameterization and DNS resolves these explicitly. DNS at atmospheric viscosity is computationally unfeasible, but it is not always necessary to resolve all scales as many flow characteristics become independent of the Reynolds number at much larger values for the viscosity than that of the atmosphere. In this paper, we build on this property (see Appendix A). One could also refer to this technique as LES with a constant eddy viscosity."*

*In this manuscript we presented a DNS with a viscosity much higher than the atmosphere to simulate a flow with moderate Reynolds numbers. The essential quality of our simulation is that we have demonstrated that the viscosity is sufficiently large to show Reynolds number independence for the statistics that we are interested in. This allows us to do less computationally expensive simulations and obtain similar results. The terminology of the simulations we performed in our research is not trivial, since the modelling techniques we used are novel. We do not claim to provide the ultimate solution but we used this technique to get better insight into turbulence above complex terrain. Further research should show which technique performs best. We now choose the terminology DNS with constant eddy viscosity in the manuscript (L140-141), however we appreciate some guidance of the editor on what is preferred from the perspective of The Cryosphere.*

*We included in the old manuscript a thorough discussion about the use of DNS and LES to cover this topic (Section Suitability of DNS). In that Section we elaborated also on why we think a LES comparison does not add value, due to the inappropriate surface model under high Reynolds numbers in complex terrain. This means comparison to LES won't give more insight in turbulent flows than DNS with a constant eddy viscosity.*

*The reviewer would like to see a comparison with field measurements, however we already did this in the old manuscript. We have compared the DNS simulations to unique Eddy Covariance measurements on Lirung Glacier in the Results section (Figure 8). Those show for the location of the EC-tower the simulated and measured sensible and latent heat flux aggregated over 1 hour for fair comparison. As the measured spatial surface temperature is input data for the model we can not use this for validation of the simulations. The comparison with the EC-measurements show acceptable agreement with the simulations, demonstrating MicroHH captures important processes.*

*For the further improvements of our analysis and presentation we refer to our replies to the reviewer's specific comments below*

Some minor comments:
Line 97, page 3. "using a novel DNS mode", please be more specific in terms of what novel algorithms or techniques are used in the DNS solver?

> - *While the code is relatively new, the numerical methods are established finite difference methods (Morinishi et al., 1998). We will remove the qualification "novel".*

Page 8. I suggest the authors add a schematic to illustrate the boundary conditions and especially, the immersed boundary method used to represent the DEM. Also, please be more specific why did the authors use 0.2 mˆ2 sˆ-1 as the eddy viscosity, as opposed to other possible values?

> *In this paper we want to focus on the application and usability of turbulence resolving models rather than discuss the model technical details. We therefore give all user-specific information and model input needed to continue with this study. We included only a limited description of the model in the methods, since this information (including the immersed boundary method) can easily be found in the main paper of the MicroHH model (van Heerwaarden et al., 2017). We think the cryospheric reader would be mainly interested in the applicability of this study.*

> *We include an explanation about the eddy viscosity term in L208: "We used a constant eddy viscosity of 0.2 $m^2s^{-1}$. Preferably the viscosity of the atmosphere is used in the simulations, however this is computationally unfeasible in our simulations. Therefore we chose the lowest possible eddy viscosity and checked the results for convergence."*

---

## Author Comment (AC2) · 3 Feb 2020

**Reply to referee 2**

Overview:
This is an important study that sheds light on the small scale variations in turbulent heat fluxes across the surface of a debris covered glacier using a high-resolution computational fluid dynamics model applied to the near surface atmosphere. It uses a novel approach to provide valuable insight into the relative importance of key, typically measured meteorological variables and, I believe, will be of great interest to the debris covered glacier scientific community (particularly to distributed modelers). The author/s clearly has/ve a solid understanding of the physical processes underlying the stated observations in the simulation results. I think this strength could be highlighted more by, in a renamed Results and Discussion section, explaining the physical processes first, then how the results demonstrate them.
The authors show that variations in melt over the surface of a debris covered glacier is due not only to debris thickness but also to variations in turbulent fluxes. Perhaps this is intuitive, but this study is the first to show it by simulating spatial patterns in wind, humidity, and temperature. This paper shows that turbulent fluxes are important to understanding the development of ice cliffs, which have previously been shown to be ablation hotspots. It importantly shows that surface heterogeneities are an important driver of energy exchange.

*We thank the anonymous reviewer for their time, and their insightful and helpful suggestions to make the manuscript more accessible and accurate.*

*We agree with the reviewer the manuscript would benefit from a renamed Results and discussion section (Section 3). We now implemented this in the new version of the manuscript. We also made some changes regarding the explanation and inserted a paragraph about the purpose of the experiments (see below at the specific comment)*

*For the further improvements of our analysis and presentation we refer to our replies to the reviewer's specific comments below*

General comments:
The discussion of the study's focus on micro meteorological variables must be honed. Turbulent fluxes are determined by wind speed and surface roughness length, as well as by temperature (sensible) and humidity (latent). In the abstract, the manuscript reads "turbulent fluxes, wind fields, moisture and temperature..."; in the conclusion, the manuscript reads "turbulent fluxes, wind fields, surface specific humidity and temperature for a debris-covered glacier is investigated." I suggest reframing the language around the purpose of this study: specifically, not listing (and, thereby, implying) turbulent fluxes as separate from wind, humidity, and temperature.

*Turbulent fluxes are indeed depending on the interplay between the wind, moisture and temperature of a specific site. We did not mean to suggest the effects are separate, however we stated the listing since we can only investigate the individual effects in our simulations.*

*We adjusted "turbulent fluxes, wind fields, moisture and temperature..." in the abstract (L14-16) and introduction (L137-141) to: In this study, we assess the effect of surface properties of debris on the spatial distribution of micro meteorological variables, such as wind fields, moisture and temperature by sensitivity tests. Subsequently we investigated how those drive the turbulent fluxes and eventually the conductive heat flux for a debris-covered glacier.*

This manuscript needs English editing (grammar and punctuation) beyond what can be provided in my review; I made some suggestions, but the manuscript needs major editing for readability. The English hampered my comprehension of the scientific basis of the paper.

*We carefully checked the grammar throughout the manuscript and made some improvements. We also incorporated the limited comments as suggested by reviewer 1. We hope the last mistakes will be noticed during proofreading and final typesetting.*

I found the Introduction especially confusing to follow–partly because of wording choices (e.g. however, thus, and nonetheless in a single sentence) and partly because there is insufficient detail on key elements of an introduction but superfluous detail on non-essential inclusions (e.g. methods and wall modeling). The

authors do not describe LES and DNS beyond spelling out the acronyms, and the authors do not discuss the reasoning behind a spatial resolution of ~1 m. The section needs clearer language to communicate a revised structure of problem/question –> hypothesis –> aims (generally exploring turbulent fluxes) –> objectives (specific, describing methods). I find that the statements of the problem (incomplete understanding of the drivers of heterogeneous melt patterns) and hypothesis are roundabout and unclear. In the last paragraph of the introduction, the aim and objectives are intermixed.

> *We are not sure which key elements the reviewer misses. We expanded our explanation about LES and DNS starting in L96, see reviewer 1. The reason we included quite some details in the method section is that people who want to continue with this study have a complete overview how to run the model MicroHH. In the Methods Section only an overview of the governing equations of MicroHH are given. We think this should give the reader enough basic information on the modelling framework. For more information we refer in the manuscript to van Heerwaarden et al. (2017).*

> *We restructured the problem description in the introduction to:*
> *"The drivers of heterogeneous melt patterns on debris-covered glaciers and the role turbulent fluxes play are not well understood. In this study, the impact of surface properties (roughness, surface temperature and surface moisture) of debris on the spatial distribution of small-scale meteorological variables, such as the turbulent fluxes, wind fields, moisture and temperature is investigated for the Lirung Glacier (Nepal) using a novel quasi DNS model with a spatial resolution of ~1 m. Observational data are used as boundary conditions, which include a high-resolution DEM (digital elevation map) and thermal imagery, retrieved from UAV (unmanned aerial vehicle) flights. We show the impact of heterogeneous surface conditions and we show that turbulent fluxes are an important contributor to the energy balance of ice cliffs. This is the first high-resolution study for a debris-covered glacier that investigates the effects of debris on meteorological variables using a turbulent fluxes resolving model. This study improves our process-understanding of debris-glacier melt, and eventually the understanding of the contribution of debris glacier melt to the current river discharge and how this will change in future. "*

> *~1m resolution is indeed not directly justified in the text. We clarify this now better in L213-215 and added:*
> *"A spatial resolution of 1 meter is the highest possible spatial resolution, given the constraints of computational power and spatial resolution of the input data. In Appendix A we show the spatial resolution of 1 meter is sufficient to capture the characteristics of the flow and that increasing the resolution will not add more information."*

Simulations: I think the suite of simulations provides valuable insight into the different variables in the energy balance. However, the author needs to make a distinction between humidity moisture and surface roughness topography/DEM as well as improve the explanation for the source of the distributed temperature and humidity data. A priori, it seems that it could be useful to conduct simulations with halved temperature and humidity deviations from the means. I think that the justification and explanation for choosing the 7 simulations needs to be strengthened and clarified, as I miss the reasons for performing the specific simulations.

> *We are not sure what the reviewer means by "the author needs to make a distinction between humidity moisture and surface roughness topography/DEM", as we discuss all experiments separately in the manuscript. In those experiments we make a clear distinction between the effects of temperature, specific moisture and DEM effects, and in the last experiment we combine those. We are sorry for the confusion but the experiments are listed in Table 1:*

*Table 1: Overview of experiments done with MicroHH. The DEM indicates the boundary condition used for the topography (0 means no DEM, ½ DEM is the original DEM halved in height, real is the spatial measured value), $T_s$ the surface potential temperature (313.3K is a homogeneous value, real is the spatial measured value), $q_s$ is the surface specific humidity (8.6 g $kg^{-1}$ is a homogeneously value, the choice for the relative humidity range is described in Sect. 2.4)*

| Experiment | Description | DEM | $T_s$ | $q_s$ |
|---|---|---|---|---|
| $HOM_{flat}$ | Homogeneous glacier | 0 | 313.3 K | 8.6 g $kg^{-1}$ |
| $HOM_{1/2DEM}$ | ½ DEM | ½ DEM | 313.3K | 8.6 g $kg^{-1}$ |
| $HOM_{DEM}$ | Roughness effects | Real | 313.3 K | 8.6 g $kg^{-1}$ |
| $HET_T$ | $T_s$ effects 'normal' | Real | real | 8.6 g $kg^{-1}$ |
| $HET_{qdry}$ | $q_s$ dry | Real | 313.3 K | Spatially RH=70-75% |
| $HET_{qmoist}$ | $q_s$ wet | Real | 313.3 K | Spatially RH=70-85% |
| REAL | "Reality" | real | real | Spatially RH=70-85% |

*For completeness we now inserted a small paragraph in L257 to make the purpose of the experiments more clear:*

*"With the $HOM_{flat}$ $HOM_{1/2DEM}$ and $HOM_{DEM}$ experiments we quantify the sensitivity of the turbulent fluxes to the topography. The $HET_T$ experiment will reveal the effects of a spatially variable surface temperature compared to a homogeneous surface temperature ($HOM_{DEM}$). The $HET_{qdry}$ and $HET_{qmoist}$ experiments are aimed to reveal the influence of heterogeneous surface specific humidity compared to a homogeneous value ($HOM_{DEM}$) and will show the effects between a relatively dry and moist debris layer. In the REAL experiment all effects are combined and will be compared to $HOM_{flat}$ and $HOM_{DEM}$ to understand the combined effects as well."*

*Also, we added in L250: "These experiments are chosen to determine the separate effects of topography, surface temperature and surface specific humidity on the surface energy balance of a debris-covered glacier."*

*We designed the experiments carefully to show the impact of roughness, humidity and temperature on the atmosphere. We would have liked to perform additional experiments but they are computationally very expensive. To our opinion these experiments reflect the most optimal configuration with the available resources. Nonetheless, experiments suggested by the reviewer could be very interesting for upcoming studies.*

Specific comments:
- Title: suggest simplifying to "using 3D turbulence-resolving simulations to investigate the energy balance of a debris-covered glacier"

*We would like to point out in the title that we specifically look at surface properties, as there are also other variables could play an important role in turbulent-resolving simulations (temperature lapse rates, wind profiles, pressure differences etc. We therefore choose to keep the existing title of the manuscript.*

- Abstract: needs to be original and not contain exact sentences from the body of the manuscript. It would be appropriate to mention that you designed a series of simulations that differed in input parameters in order to isolate and investigate the effects of varying those parameters.

*We changed the first part in the abstract to:*

*"Debris-covered glaciers account for almost one fifth of the total glacier ice volume in High Mountain Asia, however their contribution to the total glacier melt remains uncertain and the drivers controlling this melt are still largely unknown. Debris influences the thermal properties (e.g. albedo, thermal conductivity, roughness) of the glacier surface and thus the surface energy balance and glacier melt. In this study, we assess the effect of surface properties of debris on the spatial distribution of micro meteorological variables, such as wind fields, moisture and temperature by sensitivity tests. Subsequently we investigated how those drive the turbulent fluxes and eventually the conductive heat flux for a debris-covered glacier.*

- Line 12: remove first "total"
    *We implemented this change*
- Line 20: suggest replace "ascertain" with "provide insight into"

We changes this

- Line 32: explain/rephrase "non-saturated surface" or provide a reference.

*We rephrased "non-saturated surface" to "unsaturated surface"*

- Line 43: add a citation for gravel. Be more specific that you are talking about surface roughness lengths. Later, you use surface roughness interchangeably with topography (and DEM). Specifying length here would eliminate subsequent confusion.

*The Miles et al., 2017 citation is also valid for the gravel statement.*

*We changed "surface roughness" in L60 to "surface roughness length". In the manuscript we refer to "surface roughness length" as $z_0$. And "surface roughness" as the topography roughness.*

- Line 51: what does "spatial melt" mean? You don't cite any distributed energy balance model on debris covered glaciers: e.g., Reid et al (2012), Fyffe et al (2014)

*Sorry for the confusion, we meant 'model melt spatially' instead of spatial melt. We didn't included the references Reid et al (2012 and Fyffe et al (2014)), since those are based on Reid 2010 (which we cite).*

- Line 54: "we" or "they"?

*We changed it to 'they'*

- Line 55: there are many remotely-sensed observations. If excluding these, be specific. Also, missing references with data: Vincent et al (2016), Nicholson et al (2018), Nicholson Mertes (2017)

*We agree there are many remotely-sensed observation studies of the surface of debris-covered glaciers, however those do not give insight in small-scale atmospheric processes. We understand the confusion and we changed "as observations on debris-covered tongues are limited" to "as atmospheric field observations on debris-covered tongues are limited"*

- Line 58: this is a bold claim. Be specific for what observations are over short time spans.

*We added "field" in L55 to clarify we mean field observations and not remotely sensed observations.*

- Line 60: modeling would lend significant new insights, but you haven't argued it is "essential." Language implies it is the only method to shed light on the question, whereas it is only one approach.

*We rephrased L77-78 from:*

*"Due to this large spatial variation, a high-resolution modelling approach is therefore essential to capture the coupling and interaction between the surface and the atmosphere with sufficient accuracy (Mott et al., 2014)"*

*to:*

*"Due to this large spatial variation, a high-resolution modelling approach can give important new insights into the coupling and interaction between the surface and the atmosphere with sufficient accuracy (Mott et al., 2014)"*

- Line 64: "heat fluxES"

*We have changed this*

- Line 65: "gradientS"

*We have changed this*

- Line 65: Steiner et al (2018) found that bulk methods overestimate turbulent heat fluxes. . . seems relevant to mention.

*We have added in L92 "As a result, Steiner et al., (2018) found for example that bulk methods overestimate turbulent heat fluxes."*

- Line 66: summarize the "many assumptions" since this point is central to the problem you aim to address

*We rephrased L91-92from:*

*"On debris-covered glaciers however, many assumptions of the bulk-method do not hold due to the high spatial heterogeneity of atmospheric variables, a general lack of atmospheric stability or inappropriate parameterizations for the complex interaction between the heating surface and the boundary layer (Steiner et al., 2018)."*

*"However, the bulk method assumes atmospheric stability and a constant surface roughness, which are not valid over debris-covered glacier surfaces (Steiner et al., 2018)."*

- Line 70: "therefore" is for results, not for clarification. Suggest id est here. "and therefore wind, humidity and temperature fields" –> "(i.e. its wind, humidity and temperature fields)"

*OK, we changed this*

- Line 86: "we are converging to that range in this study": meaning unclear

*We do not resolve all the scales in the atmosphere in this study, however, only a marginal part of the total variance is missed and the most relevant results are independent of the Reynolds number.*

*We have added in L119: "as we show that only a marginal part of the total variance is missed and the most relevant results are independent of the Reynolds number".*

- Line 105: inconsistent formatting
    *We changed the formatting of the header.*
- Lines 107 - 108: rephrase sentence
    *We changed this line from: "The Langtang catchment has an area of approximately 560 km$^2$ and is glacierized for 30% and 25% of all glaciers is debris-covered"*
    *to:*
    *"The Langtang catchment has an area of approximately 560 km$^2$ and is glacierized for 30%. One fourth of all glaciers is debris-covered."*
- Figure 1 and most subsequent figures: include axis units and labels!
    *We added now in all spatial figures the x and y direction and its unit (m). Changes are done in all figures, except Figure 4, 8 (only title change), 10 (added label of ice cliff), A2 and A3.*
- Line 131: suggest section title "field measurements." Section as a whole needs tightening of language to be more to-the-point. It is difficult to decipher meaning.
    *We changed the section title to "field measurements"*
- Line 135: what is the purpose of this citation?
    *The data is used before in Steiner et al., 2018. For more details you can consult that paper.*
- Line 149: what is the purpose of these citations? Consider adding the words "following" or "after" if that's what you mean
    *The data is used and processed in Immerzeel et al. 2014 and Kraaijenbrink et al., 2016.*
- Lines 151 - 152: info in sentence "we only. . . the model" needs to be added to the previous section to explain the extent of the microHH domain
    *We discuss Figure 2 in L212, and we think L217-220 do fit at current place*
- Line 157: if this dash is to indicate negative, make sure it is on the same line (and page!) as the following number... and that Fig2A has the stated range included in its colorbar.
    *We adjusted the topography range in the text to 0-57m, to avoid confusion. The topography showed in Figure 2A is the model input.*
- Line 159: suggest rewrite "2- Line 162: suggest replace "used... LES)" with "a computational fluid dynamics model designed to simulate turbulent flows in the atmosphere through direct numerical simulation (DNS) and large eddy simulation (LES)."
    *We rephrased L225 to "In total 2% of the domain is covered…"*
    *L162 is replaced as suggested*
- Line 164: suggest replacing "what could be interpreted as" with "which effectively renders it"
    *We replaced this as suggested*
- Line 166: refer the reader?
    *We kept the sentence as "For specific details of the model we refer to Heerwaarden et al. (2017) but we do give a brief description of the model below."*
- Line 170: instead of what, which (error appears many times)
    *Check all cases throughout manuscript*
- Line 171: instead of therefore, thereby
    *We changed this.*
- Line 173: instead of are, is
    *We kept are*
- Lines 193 - 194: these lines need review with respect to units and consistency. (Density is not kg/kg; what is "thermal diffusivity for heat" with a value of 0.1 m$^2$/s? If you mean thermal diffusivity of water, give a calculation with specified T and P. Should be ~0.1 x 10$^{-7}$ m$^2$/s)
    *Density is in units of kg/m3, whereas the thermal diffusivity of heat in the atmosphere is chosen to be 0.1 m2/s in order to constrain the Reynolds number. In order to convert the flux to actual fluxes in the atmosphere, we take into account the actual thermal diffusivity of air (~1 e-5 m2/s).*
- Line 196: "accumulated temperature" is not intuitive. Please explain.
    *We added in the text "of a gridcell" and "and should be divided by the surface area of that cell to obtain the flux in W/m$^2$"*
- Line 209: here and elsewhere, meterS when more than 1

> *We changed this throughout the manuscript.*

- Line 210: condition is
> *We changed this*

- Line 232: themselves
> *We changed this*

- Line 233: suggest "are periodic, such that air flowing out of one side of the domain will enter on the opposite side."
> *We changed this*

- Line 249: the table lists seven experiments, not six
> We changed 6 → 7

- Lines 263 - 264: these sentences are superfluous
> We think this information can be relevant who wants to use this model as it gives an indication
> how expensive the model runs are.

- Line 265: By this point, I am missing an in-depth description of experiment design and what question each experiment was designed to answer. A reader can possibly deduce this from the results, but the purpose should be stated more explicitly.
> *We inserted a small paragraph in L349:*
> *"With the $HOM_{flat}$ $HOM_{1/2DEM}$ and $HOM_{DEM}$ experiments we quantify the sensitivity of the turbulent fluxes to the topography. The $HET_T$ experiment will reveal the effects of a spatially variable surface temperature compared to a homogeneous surface temperature ($HOM_{DEM}$). The $HET_{qdry}$ and $HET_{qmoist}$ experiments are aimed to reveal the influence of heterogeneous surface specific humidity compared to a homogeneous value ($HOM_{DEM}$) and will show the effects between a relatively dry and moist debris layer. In the REAL experiment all effects are combined and will be compared to $HOM_{flat}$ and $HOM_{DEM}$ to understand the combined effects as well."*

- Table 1 caption needs proofreading (homogenous or homogenously?; should be "spatially varying")
> We changed the caption to:
> *Table 1: Overview of experiments done with MicroHH. The DEM indicates the boundary condition used for the topography (0 means no topography, 1/2 DEM is the original topography halved in height, real is the spatially measured value), Ts is the surface potential temperature (313.3K is a homogeneous value, real is the spatially measured value), qs is the surface specific humidity (8.6 g kg-1 is a homogeneous value, the choice for the relative humidity range is described in Sect. 2.4).*

- Line 273: 2012 (not 2013)
> *We changed this*

- Line 300: suggest "all fluxes are defined as positive towards the surface except for the conductive heat flux"
> *We changed this*

- Line 302: suggest renaming the section "results and discussion" and including more discussion rather than assuming that the reader can deduce the significance from the figures (e.g. line 308: "the effect of the surface roughness on the SHF and LHF is evident (figure 3A - F)"). This section would benefit from an overview of the fact that the authors perform seven experiments designed to very key parameters that control turbulent heat fluxes in order to investigate the relative importance of various controls. Then, the subsections and figure captions could be strengthened by statements of which tests were designed to test which variables.
> *We included now the purpose of each experiment in Line 349-354 (see above) and think this is sufficient for the reader to understand the purpose of each experiment. We also renamed the section to results and discussion. We added:*
>
> *L405: "seven experiments are performed (Table 1) where key parameters are varied that control turbulent heat fluxes in order to investigate the relative importance of topography, humidity and surface temperature."*
>
> *L308 (old line number): We do not agree, since we explain what is evident from the figures in L308-L310 in lines directly following this statement: "The turbulent fluxes are intensified with increasing variability in topography, since increasing the surface roughness is directly related to the surface roughness length and the generation of turbulence."*

- Line 306: introduce that for temperature you contrast $HET_T$ with $HOM_{DEM}$ and mention how you plan to incorporate "REAL"

*See above (L349-354)*

- Figure 4: this figure summarizes the results of the experiments very well. The caption could benefit from a reminder of the sign convention for fluxes.

*We added to the caption of Figure 4: " Fluxes pointed to the surface are positive."*

- Section 3.1.4: it is not clear why only three of the experiments are discussed in this section. The last sentence of this section raises a very important point, which should be discussed further.

*We discuss in this section when homogeneous surface conditions are reasonable and when not. The last sentence is therefore a conclusion of the paragraph and does in our opinion not need more explanation.*

- Line 351: "spatial variations in"

*We changed this*

- Figures 5, 6, 7: what kind of cross sections are these? (Reference the black lines in figure 2 in the manuscript text around figures 5 - 7.)

*We added x, y and z labels in the panels of Figures 5-7, so it is now clear which cross sections are meant from Figure 2.*

- Line 360: use topography rather than DEM. The two are not interchangeable

*We have changed this.*

- Line 364: "ejections." What about diffusion and advection?

*We have added "mainly" to indicate there are other processes present but not leading.*

- Line 377: "reduce," but can't it also increase? Suggest "alter."

*We changed this.*

- Figure 7: the last column of some figures is striking because the change with height (cold to warm versus warm to cold) differs between experiments. Must discuss this and other features of the figures.

*The heterogeneous surface temperature causes negative surface temperatures on ice cliffs. We have now added in L481: "Heterogeneous surface temperatures allow negative surface temperatures at ice cliffs, resulting in reversed temperature gradients close to the surface ($HET_T$ and REAL; Figure 7)."*

- Line 372: point out specific features in figures

*We have added: " This is for example visible above the ice cliff (x=510-600m), where the wind gradient is decreasing to the bottom. "*

- Lines 395 and 402: LC or LT?

*We changed this to LT*

- Figure 8: dry debris and wet debris (change word order). It would be more intuitive to group the dried debris as A and B, the wet debris as C and D, etc.

*We changed the word order in the figure header. We have kept the numbering, as we want the figure to be 'horizontal'*

- Lines 395 - 396: this sentence seems to negate the importance of the following figure. Needs clarification.

*We have added in L515: "This means the variability of turbulent fluxes is large, even with constant surface boundary conditions."*

- Line 405: paraphrase "REAL case" for clarity

*We changed "case" to "experiment"*

- Line 407: "spatial distribution of surface moisture. . ."

*We have changed this*

- Line 410: good insight into physical properties. Not clear how to disentangle effects of topography from effects of debris.

*We compare the $HOM_{DEM}$ experiment with the REAL experiment, so in both are the topography effects and do not need to get disentangled.*

- Lines 411 - 413: this is an important point which is difficult to discern because of the language. Rewrite. Also, author(s) need to distinguish their contributions from physical principles. SHF more sensitive to T for dry debris is true b/c water has a higher heat capacity than air. "10 times as high" in next line is something learned through the model.

*We changed the sentence from:*

*Different sensitivities of turbulent fluxes to surface temperature or moisture are thus applicable in wet and dry climates and the choice of surface boundary conditions should be chosen carefully for simulations.*

*To:*

*"Turbulent fluxes have different sensitivities to surface temperature and moisture indicating that the sensitivities are different in wet and dry climates. As a result surface boundary conditions should be chosen carefully for simulations."*

> *In L535 it is stated that the LHF is 10 times as high in the experiments, so it is clear this is learned from the model and is not used as a physical explanation.*

- Lines 416 - 417: If these are averages, why is there an uncertainty? And why is it greater than the average?

> *It is not uncertainty but the standard deviation of spatial variability; we have now indicated this in L539:*
> *"Domain averaged (with its spatial standard deviation)"*

- Line 421: "near-surface air is saturated" contradicted at end of paragraph

> *We mean only the air above the ice cliff, at the end of the paragraph we discuss the whole domain. The relatively dry eddies generated above (dry) debris flow over a saturated ice cliff. This is therefore not a contradiction with L424-425.*

- Line 429: weighted how? Approach needs an explanation.

> *The AWS data are presented as a weighted average everywhere in manuscript. We stated this in L201*

- Line 441: Suggest "figure 9 shows conductive flux into the debris under the seven simulations."

> *We changed this.*

- Line 452: can you say anything here about the physics with respect to thermal conductivity, density, and heat capacity? Conductive heat flux is determined by the temperature gradient in the debris, so this is expected. Clarify what additional insight your simulations provide.

> *We define the conductive heat flux as the surplus of the surface energy balance (equation 12). To clarify this, we have added in the manuscript: "The energy reaching the ice below the debris is dependent on the thermal conductivity, density and heat capacity of the debris."*

- Table 2 line 480: what is the significance of the non-normal distributions, for which standard deviations exceed the means? Consider using other statistical metrics instead/also.

> *It denotes the spatial variation in the domain, we have added in L620 "…heat flux (± spatial variation)…"*

- Table 2: refer to this table in the text

> *This is done in L580*

- Table 2: what is the breakdown of ice-cliff area vs. debris area in the domain? Mention in caption and discuss in text if you contrast in a table.

> *Ice cliffs cover 2% of the domain, as stated in L225. We now repeat this in the caption for completeness.*

Lines 467 - 472: relate to recently published findings on ice cliffs in HMA

> *We have added: "Other studies found ice cliffs melt between 5.7 and 13.7 times faster than ice below surrounding debris debris (Brun et al., 2016; Sakai et al., 2002; Buri et al., 2016; Reid and Brock, 2014). Although those consider total melt rather than the surface energy balance, the pronounced differences between melt on ice cliffs and debris are in line with our research."*

- Line 477: observationS of what value(s)?

> *Observations of specific humidity, as stated at start of this line*

- Lines 504 - 505: clarify with "every 10 seconds of [time interval] in the simulation"

> *We have added "time" to indicate panels are shown with 10 seconds time interval*

- Figure 10: label ice cliff face, give variables in headings for each row of figures (A - F, G - L, and M - R), and specify the exact time interval in the caption. Also, should be 660<x<500.

> *The ice cliff is now made more clear in Figure 10. The labels are given in the individual subplots. We changed 660<x>500m to 660<x<500m. The exact time intervals are given in the caption. We added 'with 10s time interval'.*

- Line 529: "and is" to "that was"

> *We have changed this.*

- Line 531: turbulent fluxes likely play

> *We have changed this.*

- Line 532: it is appropriate to refer to the figure, but the reader cannot "see" the windspeed derivative where the ice cliff changes slope. Labeling the ice cliff and circling the region of interest on the figure would help.

> *We agree the reader can not see the derivative, however we focus on the vertical wind speed variations; those can be seen in Figure 5.*

*We label the ice cliff now in Figure 10:*

[Figure]

- Line 535: "wind flow does flow": rewrite

> *We changed this to "the main wind flow is over the depression"*

- Section 4: suggest renaming as "sensitivity to Reynolds number" and start with a short description of the Reynolds number and why you chose to perform a sensitivity test on it. The first paragraph of this section states that both DNS and LES are impossible. The meaning of this paragraph is especially difficult to discern from the English that is used.

> *We would like to state that neither DNS nor LES is impossible, but that each has its own set of problems that need to be overcome in order to make meaningful simulations. We renamed the Section as suggested by the reviewer. We added a paragraph about the Reynolds number in L688:*
>
> *"The Reynolds number (Re) is a measure for the flow characteristics and is the ratio between inertial and viscous forces in a fluid (Eq. 18). A low Reynolds number would indicate a laminar flow and high Reynolds numbers a turbulent flow.*
>
> $$Re = \frac{uL}{v}, \qquad\qquad\qquad\qquad (18)$$
>
> *Where u is the velocity (m/s), L is the characteristic length scale (m) and v is the kinematic viscosity (m²/s) of the fluid. By decreasing the kinematic viscosity the fluid will become more turbulent and resolve smaller scales."*

- Lines 545 - 550: what is the effect of the different resolutions on the profiles near the surface, where the difference is most apparent? You show the resolution is not too large for achieving accuracy, but could the same patterns be captured with a resolution larger than one meter? How much larger?

> *We did some sensitivity tests with coarser resolutions, however those missed the details a 1m resolution simulation gives. The focus of the paper is not on the sensitivity to the spatial resolution and we decided not look in detail to this.*

- Lines 561 - 563: rephrase

> *We rephrased from: "The simulation at low viscosity naturally resolves smaller scales than the simulation with a higher viscosity, yet the additional variability does not relevantly add variance to the signal and are therefore irrelevant for the flow (Figure A3, wave numbers>11)"*
>
> *To: "The simulation at low viscosity naturally resolves smaller scales than the simulation with a*

*higher viscosity (see Equation 18). In Figure A3 we see that the additional variability does only add a small amount of variance to the signal and is therefore irrelevant for the flow (wave numbers>11)"*

- Line 556: topography, not orography

 *We changed orography to topography*

- Section 5 "Limitations": this section especially needs proofreading by the author(s). The writing makes it difficult to discern many of the concepts, which are ones important to the paper.

 *We made some changes in the Section Limitations:*
 L722: over → for
 L724: added "is unknown" end of sentence
 L724: is → was
 L725: gain understanding in →gain understanding of
 L726: show → showed
 L726: turbulence resolving → turbulence-resolving
 L727: identify → identified
 L729: hereafter → in the following paragraphs
 L730: removed 'very'
 L732-733: "Debris-covered glacier studies normally deal with this unknown by assuming the debris surface…" → "Studies normally deal with this limited information about surface moisture by assuming that"
 L757: Step forward for → step forward to
 L781: prescribed →included
 L792: lead to: result in
 L794: representative for → representative of
 L794: However→ However,
 L794: such → so
 L795: High resolution → high-resolution

- Line 576: add that debris moisture is important to not only turbulent heat fluxes but also the conductive heat flux that ultimately melts glacier ice

 *We added "and therefore the conductive heat flux that ultimately melts the ice."*

- In this discussion of moisture, can you add some discussion of season and any implications of your findings for the monsoon season in particular?

 *We added in L626: "During monsoon surface moisture will be higher than in the rest of the year, indicating the conductive heat flux at the surface will be higher during monsoon than in winter and spring"*

- Line 598: here, it sounds like the AWS station you used \*happened\* to be in a spatially-representative place but that you got lucky because "in reality it remains very hard to locate a station such that it is representative for the whole domain." If this is not the case, please change the language. Additionally, it would be helpful if you quantified the amount of bias that could be introduced by upscaling point measurements not representative of the domain.

 *Our experiments made sure the homogeneous value is representative for the domain, as this is an average of the heterogeneous values. This means our homogeneous value is per definition representative of the domain. The bias of upscaling point measurements is given by comparing homogeneous surface conditions (HOM experiments) and heterogeneous surface conditions (HET experiments).*

- Line 615 - 618: these lines read that you investigated the impact surface temperature and specific humidity have on surface specific humidity and temperature. Please rewrite with greater clarity.

 *We agree this is confusing. We changed the sentence to "In this study, the impact of surface properties of debris (surface roughness, temperature and specific humidity) on the spatial distribution of small-scale meteorological variables, such as turbulent fluxes and near-surface wind fields, specific humidity and temperature for a debris-covered glacier is investigated.*

- Line 642: I would think that that the bare ice on ice cliffs has a higher albedo than debris covered surroundings; explain or cite otherwise.

 *Ice cliffs generally have an amount of sand/dust on the ice cliffs that lowers the albedo. This is stated in Steiner et al. (2015), and indicated in the methods Section, L377.*

- Line 648: this paragraph is important to include only if you quantify and show an example of the large biases that are possible.

 *We showed the bias between homogeneous and heterogeneous variables in the results section and keep this paragraph.*

- Line 661: add labels to axes in the .gif's (videos)
       *We have done this for the vertical cross sections and conductive heat flux videos. It will take a few days before this is visible on Zenodo*
- References: cite publications in The Cryosphere, not The Cryosphere Discussions where possible (e.g., Rounce et al., 2015)
       *We update Rounce et al., (2015) to the fully published article.*